



# Modeling subgrid lake energy balance in ORCHIDEE terrestrial scheme using the FLake lake model

Bernus Anthony and Ottlé Catherine

Laboratoire des Sciences du Climat et de l'Environnement, IPSL, CEA-CNRS-Université Paris-Saclay, Orme des Merisiers, 91190 Gif-sur-Yvette, France

**Correspondence:** Anthony Bernus (anthony.bernus@lsce.ipsl.fr)

**Abstract.**

The freshwater 1-D FLake lake model was coupled to the ORCHIDEE land surface model to simulate lake energy balance at the global scale. A multi-tile approach has been chosen to allow the modelling of various types of lakes within the ORCHIDEE grid cell. The different categories have been defined according to lake depth which is the most influential parameter of FLake, but other properties could be considered in the future. Several depth parameterization strategies have been compared, differing
by the way to aggregate the depth of the subgrid lakes, i.e., arithmetical, geometrical, harmonical mean and median. Five atmospheric reanalysis datasets available at $0.5°$ or $0.25°$ resolution, have been used to force the model and assess model systematic errors. Simulations have been performed, evaluated and intercompared against observations of lake water temperatures provided by the GloboLakes database over about 1000 lakes and ice phenology derived from the Global Lake and River Ice
Phenology database.

The results highlighted the large impact of the atmospheric forcing on the lake energy budget simulations and the improvements brought by the highest resolution products (ERA5 and E2OFD). The median of the Root Square Mean Errors (RMSE) calculated at global scale range between 3.2 K and 2.7 K among the forcings, CRUJRA and ERA5 leading respectively to the best and worst results. Depth parameterization strategy appeared to be less influent, with RMSE differences less than 0.1 K for
the four aggregation scenarios tested.

The simulation of ice phenology presented systematic errors whatever the forcing used and the depth parameterization. Freezing onset was shown to be the less sensitive to forcing and depth parameterization with median of the errors ranging between 10 and 14 days. Larger errors were observed on the simulation of the end of the freezing period significantly influenced by the atmospheric forcing used. Such errors already highlighted in previous works, could be the result of deficiencies in the
modeling of snow/ice parameterization processes. Various pathways are drawn to improve the model results, including the use of remote sensing data to better constrain the lake radiative parameters (albedo and extinction coefficient) as well as the lake depth thanks to the recent and forthcoming high resolution satellite missions.





# 1 Introduction

Lakes are an important component of the continental hydrology and interfere in the global carbon cycle. Lakes do indeed
play a role in many physical and biogeochemical processes at the land-atmosphere interface such as the transfers of energy,
water, carbon and other greenhouse gas (GHG) like methane. However, since they represent only a small percentage of the land
surface (about 3.7% as shown by Verpoorter et al. (2014)), their contribution at the global scale is small (at least regarding the
energy and water budgets). Therefore, lakes started to be explicitly represented in Earth System Models (ESMs) only recently,
thanks to the increased grid resolutions and computer advances.

Lake distribution is spatially unequal all other the world with two regions in the high latitudes (Canada and Scandinavia)
showing the highest number of lakes (Downing et al., 2006). The second characteristic of this distribution is the presence of
large lakes in Northern America and Africa which represent 27% of the lake water volume and 11% of the lake surface (Messager et al., 2016). Their role on the physical properties of the low atmosphere and the importance to model these processes,
has been demonstrated by many studies. For example, Bonan (1995) showed that the implementation of lakes in the NCAR-
CCM2 climate model, lead to an increase of the latent heat fluxes transferred to the atmosphere and to a decrease of the annual
amplitude of the surface temperature. They found that in the high inland lake regions in July, the average surface temperature
was cooler by $2°C$ to $3°C$ and that the latent heat fluxes increased by 10 to 45 $W/m^2$ depending on the region.

Lakes play also a buffer role in the hydrological cycle impacting substantially the river regimes. As an example, in Alaska,
Bowling and Lettenmaier (2010) explain the reduction of the spring peak flow of the Putuligayuk river by the lake storage
which represents up to 80% of the snowmelt.

Regarding the carbon and nitrogen cycles, lakes emissions are poorly constrained but it is recognized that their contribution
is significant. Lakes are a carbon source for the atmosphere, with emissions of about 110 TgC of $CO_2$ in annual mean, but
they are also a sink since an important part of organic carbon is buried into the bottom sediments (Cole et al., 2007). Another
important GHG is the methane produced by nitrogen oxidation, processes which are presently poorly understood but could
lead to lake emissions between 8 and $48\,Tg\,C - CH_4\,y^{-1}$ , comparable to the emissions of the oceans (Bastviken et al.,
2004). For example, West et al. (2016) estimated that, between 8% and 16% of the non-anthropic methane emissions are
coming from lakes. Recently, Beaulieu et al. (2019) gave a higher estimation of lake methane emissions with an amount of
$112\,Tg\,C - CH_4\,y^{-1}$.

Given the role of lakes on the global carbon and water cycles, the large uncertainties related to the GHGs budget (Saunois
et al., 2020) and the increasing spatial resolution of ESMs allowing to better account for the land surface diversity, it appears
necessary to implement water bodies in ESMs. This implementation requires as a first step to solve the surface energy balance
in order to be able to predict the time evolution of the surface temperature and fluxes exchanged with the atmosphere. Among
all the models developed for that purpose, the one-dimensional approaches are the most used in ESMs because of their low
computational cost and the robustness of the parameterizations with a little number of free parameters. In this category, the
FLake model (Mironov, 2008) and the Hostletler model (Hostetler and Bartlein, 1990) are the most used to represent lakes
in ESMs. These two models follow different approaches to solve the lake energy budget and to calculate the energy vertical





transfers and the water temperature profile: in FLake, the temperature profile is computed through the self-similarity theory whereas in Hostleter's model, the eddy diffusive equations are explicitly resolved. An intermediate methodology developed by MacKay (2012) resolving the turbulence in the mixing layer with a bulk approach may also be mentioned. These three

models have shown good performances when compared at site level in various intercomparison exercices such as LakeMIP (Stepanenko et al., 2010, 2013).

As such, FLake has been implemented in numerous climate and Numerical Weather Prediction Models (NWP). For example, it has been coupled to the German regional atmospheric model (COSMO) and its TERRA land component (Mironov et al., 2010), the European ECMWF atmospheric forecasting model and its HTESSEL land surface component (Balsamo et al., 2012;

Dutra et al., 2010), the UK Met Office Unified Model and its JULES Land simulator (Rooney and Bornemann, 2013), and the French CNRM-CM5 ESM and its SURFEX interface (Le Moigne et al., 2016). The coupling of FLake with all these LSMs showed strong benefits in the climate and weather simulations. In addition, the LISS model based on the Hostetler model is used in the CLM land component of the CESM climatic model developed by the NCAR institute (Oleson et al., 2013) and in the ELM land model based on the same core of CESM as BCC-AVIM, (Golaz et al., 2019) and NorESM2 (Seland et al.,

2020). The LISS model is also implemented in LM4 which is the land component of the GFDL ESM (Milly et al., 2014) and in the land component CLASS of the Canadian ESM (Martynov et al., 2012). More recently, mass balance equations have been implemented in some ESMs in order to solve the coupled energy and water cycles. For example, Huziy and Sushama (2017) implemented lake mass budget in the CLASS LSM and tested it for regional applications. Besides, Guinaldo et al. (2021) developed the MLake mass budget model in SURFEX, thus enabling to improve the global scale water discharges and

the seasonal evolution of lakes water levels.

The work presented in this study aims to include lakes in the IPSL ESM and more precisely in its land surface component ORCHIDEE. Contrary to Krinner (2003) who directly introduced the Hostetler model in the atmospheric model component (LMDZ) of the ESM and by this way, highlighted the role of water surfaces on the boreal climate, we chose here to represent explicitly the lakes in ORCHIDEE and to resolve, as a first step, the energy balance. For that purpose, the FLake lake model

was coupled to ORCHIDEE and various developments have been done to assess the lakes features and set up the new parameterizations. We present these developments here and the results obtained through a series of simulations devoted to assess the respective role of the atmospheric forcing and of the depth parameterization in the model performances. We focus on the analysis of the simulated surface variables (temperature, freezing state and turbulent fluxes) which are impacting the atmospheric surface boundary layer. Model performances are analyzed at the lake scale and globally, against satellite datasets. The

contribution of lakes in the simulation of the surface temperature and energy fluxes is quantified. The following steps including further model developments and parameter optimization are finally discussed.



## 2 Data and Modeling framework

### 2.1 Datasets used

#### 2.1.1 Land Cover map

The ESA Land Cover CCI product (v2.0.7, Bontemps et al. (2015), ESA (2017) ) developed in the framework of the Climate Change Initiative program of the European Space Agency (ESA) was used to map the natural vegetation and derive the OR-CHIDEE PFTs. The product is available at a resolution of 300 m and the global maps are provided on a yearly basis, over the period (1992 – 2020). The interpretation of the 38 land cover classes in terms of ORCHIDEE PFTs is fully presented in Poulter et al. (2015) and Hartley et al. (2017), and detailed in the dedicated website https://orchidas.lsce.ipsl.fr/dev/lccci. In this work,

we used the PFT maps generated at the resolution of $0.25°$ and $0.5°$ for the standard version of ORCHIDEE.

#### 2.1.2 Lake area and depth

The HydroLAKES database (Messager et al., 2016) has been used to characterize lake fractions and depths. The dataset provides georeferenced polygons for 1.4 million lakes of size larger than 10 ha. A mean depth is provided for each water body, estimated either from in situ measurements or from geomorphological considerations allowing to assess lake shape

from which depth is derived. Additional attributes include estimates of the shoreline length, water volume and residence time but they were not used in this work. Moreover, all lakes are co-registered to the global river network of the HydroSHEDS database used in ORCHIDEE to describe the river system and constrain the water routing scheme. This specificity will insure the coherence for the following developments when lakes will be connected to the river routing. The data can be downloaded at https://hydrosheds.org/page/hydrolakes.

#### 2.1.3 Lake surface temperatures

The Global Observatory of Lake Responses to Environmental Change (GloboLakes) database is used to evaluate our simulations. This dataset has been developed thanks to a project funded by the Natural Environment Research Council (NERC) in the continuity of the ARC-Lake project (MacCallum and Merchant, 2012) funded by the European Space Agency. It provides daily observations of Lake Surface Water Temperature (LSWT) for about 1000 lakes distributed all other the world, with uncertain-

ties and quality level. The observations cover the period 1995-2016 with a spatial resolution of $0.05°$ and combine different instruments such as the AVHRR (Advanced Very High Resolution Radiometer) on MetOpA, AATSR (Advanced Along Track Scanning Radiometer) on Envisat and ATSR-1/2 (Along Track Scanning Radiometer) on ERS-1/2 (European Remote Sensing Satellite) covering different periods. The time series from ATSR-2 are available for the period 1995-2002, from AATSR for the period 2002-2010 and from AVHRR for the period 2007-2016. A common processing methodology and the fact that all the

satellites have a quasi polar orbit leading to a constant local solar time, allowed a time homogeneity in the measurements and a homogeneous resulting dataset (Carrea and Merchant, 2019). Given the lack and lower quality of the data before 2002 due to the older instruments, we only used the period 2002-2016 for the model evaluation. For the whole period between 1995 and





2009 and for the previous version of the database ARC-Lake, the product uncertainty estimated by Carrea and Merchant (2019) is less than 1 K. The comparison against local measurements show that errors are slightly larger for day time measurements

(0.7 K)) compared to night time (0.5 K) with a small negative bias (-0.2 K for day and -0.1 K for night) linked to the thermal skin cooling effect. The data can be found at: http://dx.doi.org/10.5285/76a29c5b55204b66a40308fc2ba9cdb3.

### 2.1.4 Lake ice phenology

The Global Lake and River Ice Phenology Database is used to validate the lake freezing processes. This dataset provides the start, end and duration of the freezing period for 857 lakes located in the northern latitudes. The dataset is based on in situ

measurements which are sometimes discontinuous and acquired over different periods. The data are available for some lakes back to the year 1850 (Benson, 2002). The data can be downloaded at https://doi.org/10.7265/N5W66HP8 .

### 2.1.5 Atmospheric forcings

Different atmospheric reanalysis are used to force the ORCHIDEE model and assess the model sensitivity to meteorological forcing uncertainties. These reanalysis differ by the model and data used to constrain the atmospheric fields, differences that

may introduce some biases. In this work, we used five different datasets available at three different spatial resolutions: 31km, $0.25°$ and $0.5°$ and three different temporal resolutions, hourly, 3-hourly and 6-hourly.

Then, for the simulations performed at $0.5°$ resolution, we used: a)WFDEI-CRU, a 3-hourly $0.5°$ global forcing product which is a combination of the ERA-interim reanalysis and the Climate Research Unit (CRU)-TS 3.1 climatological data (Harris et al., 2014) presented by Weedon et al. (2011), b) CRUJRA, a 6-hourly $0.5°$ global forcing product, which is based on JRA-55

reanalysis (Kobayashi et al., 2015) corrected by CRU-TS 3.2 (Harris et al., 2014), and finally c) CRUNCEP, a 6-hourly $0.5°$ global forcing product, which is a combination of two datasets: the CRU-TS 3.2 and the National Center for Environmental Prediction (NCEP) reanalysis product described by Viovy (2018). ERA5 (Hersbach et al., 2020), a hourly global forcing product at a resolution of 31 km, and E2OFD (Beck et al., 2017), a 3-hourly $0.25°$ global forcing product complete the selection. All the data have been formatted by the ORCHIDEE team to be used and interpolated at the half-hourly time step, following

the methodology presented in: https://forge.ipsl.jussieu.fr/orchidee/wiki/Documentation/Forcings. In brief, the interpolation of the meteorological variables is done linearly except for the solar radiation where the interpolation between two times, follows the typical daily evolution according to the solar zenithal angle. It should be noted that no extra atmospheric fields are needed to solve the energy balance at the lake – atmosphere interface, same as what is done for the vegetation part.





## 2.2 Modeling framework

### 2.2.1 ORCHIDEE land surface model

ORCHIDEE is the land surface component of the IPSL ESM. It simulates the surface energy balance and the water, carbon and nitrogen cycles for large scale and climate applications. The model is composed of two main modules named SECHIBA and STOMATE. SECHIBA resolves the thermal and hydrological processes at a half-hourly time step, and computes the key surface variables and fluxes such as the soil temperatures (Wang et al., 2016) and water content (de Rosnay, 2003) with a 11-layer discretization, the evolution of the snow cover with an explicit 3-layer representation (Wang et al., 2013) and the photosynthesis of the vegetation (Vuichard et al., 2019). The second module STOMATE computes with a daily time step, the fully coupled carbon and nitrogen dynamics and therefore calculates key processes such as the vegetation respiration, the soil carbon dynamics, the litter decomposition, the vegetation phenology, constrained by the nutrients availability (Vuichard et al., 2019). The model can be run in forced mode with atmospheric variables prescribed or coupled with the atmosphere simulated by the atmospheric general circulation model LMDZ. In ORCHIDEE, the land surface is represented by two Surface Functional Types (SFTs) differentiating bare and vegetated surfaces from glaciers. The first SFT allows to represent the land surface into fractions of 15 different broad Plant Functional Types (PFTs) merging both plant species and main climate zones, one of them representing bare soils. In the standard versions of ORCHIDEE, urban areas and water surfaces are not represented explicitly and are characterized by a fraction of bare soils. The fractions of PFTs assigned in a grid cell are determined by a land cover map and a cross-walking-table (CWT) allowing to link the land cover classes to the ORCHIDEE PFTs. This procedure is fully presented in Lurton et al. (2020) and more recently in Harper et al., in preparation. The model can be used at various scales, from the local to the global scale, with grid resolutions constrained by the scale of the atmospheric input fields (incoming shortwave and longwave radiations, surface air temperature and humidity, wind speed). In this work, the model version used is the AR6 version which participated to the CMIP6 model intercomparison exercice, and contributed to the IPCC $6^{th}$ Assessment Report (Boucher et al., 2020; Cheruy et al., 2020).

### 2.2.2 FLake lake model

FLake (Mironov, 2008) is a 1-D thermodynamic lake model developed for NWP purposes. It is a bulk model capable of predicting the vertical temperature structure and mixing conditions in lakes, given the meteorological conditions at the atmosphere interface (incoming radiation, air temperature and humidity, wind speed). The water temperature profile is represented by a single mixed layer with a uniform temperature above a thermocline. Extra modules are implemented to model the snow, ice and sediment profile temperatures into specific layers.

The structure of the lake thermocline is parameterised using the concept of self-similarity. Therefore, the depth-temperature relationship depends on a shape factor, which is resolved at each time step according to the boundary conditions. The same approach of "assumed shape" is used to represent the snow, ice and sediment layer temperature profiles.





The resolution of the bulk energy budgets of the mixed layer and thermocline, allows to calculate the model prognostic variables, i.e. the mixed-layer temperature and depth, the free water bottom temperature, the thermocline shape factor, the ice layer top temperature and ice thickness.

If the snow module is activated, the snow temperature at the atmosphere interface and the snow depth are calculated, based on the bulk resolution of the heat budgets of the snow layer. It should be noted that the model resolves only the energy budget

equations and that the water balance is not solved. That means that the water volume is kept constant in time, the depth and surface extent are therefore input parameters of the model. In addition, two important radiative parameters (surface albedo and light extinction coefficient) and the surface fetch involved in the calculation of the surface fluxes need to be prescribed to run the model.

FLake does not model the hypolimnion, the layer under the thermocline which is present or may appear seasonally in

stratified lakes and where the water density is the highest with a constant temperature around $4°C$. For deep lakes and to get around modeling this layer, the lake depth is set to a maximum value of 50 meters as recommended in the FLake documentation available on the dedicated website (http://www.flake.igb-berlin.de/site/download).

## 3 ORCHIDEE-FLake coupling developments

### 3.1 Integrating FLake into ORCHIDEE

The FLake model has been implemented in the ORCHIDEE model, following the same approach proposed by Salgado and Le Moigne (2010), i.e., multi tile approach where separate energy budgets are computed for the vegetation and the lake respective fractions within a grid cell. The developments were done in the SECHIBA routine driving the resolution of the water and energy surface budgets. All the tiles shared the same atmospheric forcing data and exchanges fluxes with the atmosphere according to their respective surface area. The number of lake tiles within a grid cell is variable and can be defined by the user,

in order to allow the simulation of different types of water bodies which could present different properties such as depth or radiative properties. Figure 1 presents a general scheme of the ORCHIDEE-FLake coupled model. In the following sections, we present some refinements that were brought in FLake in order to account for recent improvements in the resolution of the water temperature profile and in the snow model. The parameterization of lakes within the ORCHIDEE grid cell and the choices made for their characterization is then presented.

### 3.2 FLake model refinements

#### 3.2.1 Shape factor revision

As already noted, in FLake, the temperature profiles in the thermocline layer (but also in the sediment layer not activated in this work) are solved following the concept of self-similarity. The temperature profile in the thermocline is computed through a shape function parametrerized by an empirical polynomial equation which satisfies boundary conditions at the top and at



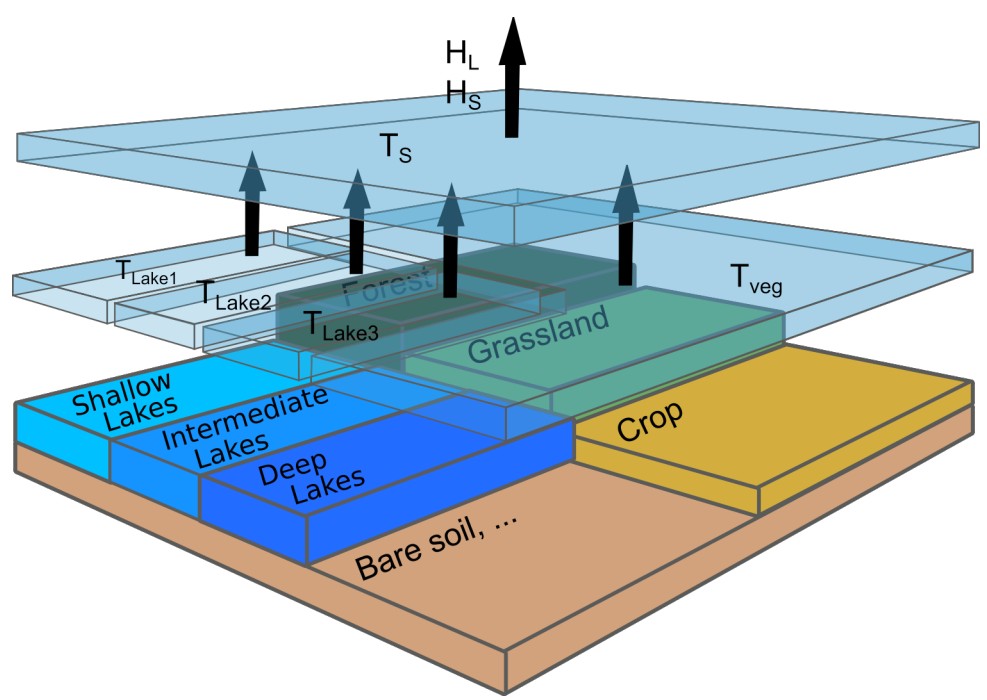

**Figure 1.** Lakes tiles representation in ORCHIDEE land surface model

the bottom of the thermocline. The equation was proposed by Malkki and Tamsalu (1985) where two cases are differentiated depending if the mixed layer $h$ is developing or retracting following Equation 1.

$$
\Phi_\Theta(\zeta) = \begin{cases} 1 - (1-\zeta)^3 & \text{if } dh/dt > 0 \\ 1 - 4(1-\zeta)^3 + 3(1-\zeta)^4 & \text{if } dh/dt \leq 0 \end{cases} \tag{1}
$$

$\Phi_\Theta$ represents the normalised temperature depending of the normalised depth $\zeta$ as follows:

$$
\begin{aligned} \Phi_\Theta &= \frac{T_s - T(z)}{T_s - T_b} \\ \zeta &= \frac{z}{D} \end{aligned} \tag{2}
$$

where we define $T_s$ as the lake surface water temperature, $T_b$ as the lake bottom temperature and $T(z)$ is the water temperature at the depth $z$.





Mironov (2008) defined a shape factor $C_\theta$ as the integration over the depth of the temperature profile. Equation 3 controls the evolution of the shape factor and allows to make the transition between the two temperature profiles. This factor linearly changes with time and proportionally to a relaxing characteristic time $t_{rc}$ between the two extreme values $C_\theta^{\min}$ and $C_\theta^{\max}$.

$$\frac{dC_\theta}{dt} = \text{sign}(dh/dt)\frac{C_\theta^{\max} - C_\theta^{\min}}{t_{rc}}, \quad C_\theta^{\min} \leq C_\theta \leq C_\theta^{\max} \qquad (3)$$


Some issues were reported by Salgado and Le Moigne (2010) who found unrealistic behavior of the temperature profile in spring seasons for long term simulations, with a too large decrease of the bottom temperature. They reveal that the numerical instabilities of the shape factor during winter was playing a role in this issue. On our side, we observed similar instabilities even when no freezing conditions were encountered. Salgado and Le Moigne (2010) proposed to limit the shape factor time
evolution to a threshold fixed to $0.01\,h^{-1}$ and to rise up the minimum value $C_\theta^{\min}$ from 0.5 to 0.65. They found that the deep temperature evolution with these changes was more realistic with a large decrease of the numerical instabilities. After some tests, we came to the same conclusions and chose to apply the same corrections to the shape factor parameterization.

### 3.2.2 Lake snow module

The snow module within FLake which allows to simulate the time evolution of the snowpack when the lake is frozen, was not
well tested in its original parameterization (Mironov et al., 2010). The snow temperature is represented following the same approach as the one developed for the ice layer below, i.e., integral heat budget of the ice and snow layers and self-similarity assumption of the temperature profiles. At the snow-ice interface, Equation 4 is used to calculate the snow temperature and derives from the condition of continuity of the heat flux.

$$T_{snow} = T_{ice} + R_h \cdot (T_{ice} - T_{freeze}) \qquad (4)$$

$R_h$ is the ratio of the thermal resistance between the ice layer and the snow layer equal to $\frac{H_S}{H_I}\frac{\Phi_I'(1)}{\Phi_S'(0)}\frac{k_I}{k_S}$ where $H_I$ and $H_S$ are the ice and snow height and $k_I$ and $k_S$ are the thermal conductivity of ice and snow. This formulation can lead to numerical instabilities for high values of $R_h$. To fix this issue, Pietikäinen et al. (2018) when coupling the German REgional MOdel (REMO, Jacob and Podzun (1997)) with FLake, proposed to set a threshold for the ice layer thickness above which the snow layer can develop. On our side, we did not implement this solution and prefer to decrease the time step of the snow module to
prevent the numerical instabilities with the advantage of preserving the water budget. Various tests were performed to adjust the time split factor which has been finally set to a value of 50. Therefore, when the snow thickness is less than 3 cm, the time step of the resolution of the snow module is decreased by a factor of 50. Other improvements to the snow module of FLake were proposed by Semmler et al. (2012). The improvements concern the parameterizations of the snow density, the thermal conductivity and the snow/ice albedos. Based on data acquired on the Bear lake in Canada, Semmler et al. (2012) revised
the snow density relationship with height which can lead to unrealistic values during the melting period and tested various parameterizations including the use of a constant value of $320\,kg \cdot m^{-3}$. Their results show the benefit of parameterizations





accounting for snow aging, but given that the use of on a constant value permitted them to better represent the snow and ice cover dynamics compared to the original FLake parameterization, we chose this option as a first step.

The original equation of the snow thermal conductivity $k_s$ was given by Equation 5 with $k_{s_{max}}$ and $k_{s_{min}}$ the maximum

and minimum values of the snow heat conductivity and $h_s$ the snow depth. It was replaced by Equation 6, where $k_s$ is set to $0.14\,W\,m^{-1}\,K^{-1}$, $k_i$ is the heat conductivity of the lake ice ($2.29\,W\,m^{-1}\,K^{-1}$) with $c$ an empirical constant ($5\,m^{-1}$).

$$k_s = min(k_{s_{max}}, k_{s_{min}} + 1.3 \cdot h_s cdot \frac{\rho_s}{\rho_w}) \tag{5}$$

$$k_{seff} = k_s + (k_i - k_s) \cdot e^{c \cdot h_s} \tag{6}$$

Following Semmler et al. (2012), Pietikäinen et al. (2018) revised also the calibration of the ice and snow albedos and

proposed to change the white/blue ice albedos values from 0.6/0.1 to 0.5/0.3 and the dry/melt snow albedo values from 0.6/0.1 to 0.87/0.77. We implemented the same modifications in our developments.

## 3.3 Model settings

ORCHIDEE requires the description of the land surface variability in each model grid cell to simulate the energy and biogeochemical processes. As already noted, the standard version uses PFT maps derived from the CCI Land cover product. Since

the standard PFT maps do not consider water surfaces, the permanent water bodies have been merged with the bare surfaces in the corresponding PFT. The new splitting in SFTs with different energy budgets calculation, requires that lake fraction is first removed from the bare soil fraction and transferred to the new lake SFT. For that purpose, new PFT maps excluding lakes for each year of the simulation have been generated. In a second step, we have worked on the definition of the lake fraction within a grid cell and the needs to split it according to the depth, following the recommendations of Bernus et al. (2021). Indeed, one

important outcome of our previous work, was to quantify the role of FLake parameters in the simulation of the surface variables and to show that depth is especially determinant in the case of shallow lakes. Therefore, we decided to prescribe the number of lake tiles in our model. This new parameter that has to be set prior to the simulation and allows to define different types of lakes within a grid cell, differing by their input parameters (depth, surface area, radiative properties, etc...). The methodology developed to define those parameters are presented in the following.

### 3.3.1 Lake surface area and depth

The HydroLAKES database (Messager et al., 2016) was analyzed to parameterize lake surface area and depth in ORCHIDEE. In the study of Bernus et al. (2021), three categories were defined according to depth ranges: shallow lakes with depth below 5 m, deep lakes with depth larger than 25 m and the intermediate class in between (depth range between 5 and 25 m).

Figure 2 shows the lake surface fractions (in logarithmic scale), plotted at $0.5°$ resolution, for the three categories of lake

depths and aggregated in a single class. The median values of the corresponding depths are shown on the same figure. The



**Figure 2.** Lake fractions (top) and lake depth (bottom) averaged for each lake categories at $0.5°$





higher amount of lake surfaces in the high northern latitudes mainly in northern Canada, Scandinavia and Western Siberia, are clearly highlighted. The lake distribution shows a minimum at the latitudes around $20° - 30°$ North and South, in coherence with the arid climates found at these latitudes, produced by the descending branch of the Hadley circulation. The plots of the corresponding depths show a higher concentration of shallow and intermediate lakes in the northern high latitudes, whereas

deep lakes look well distributed on all the continents. The major big lakes appear clearly on the plots, as the Great lakes in Canada, the Victoria lake in South Africa, the Malawi lake in South America or the Caspian sea and Baïkal lake in Eurasia.

### 3.3.2 Other lake parameters

FLake requires the prescription of other parameters like the radiative parameters of water (surface albedo and extinction coefficient) and the wind fetch. For the radiative parameters, the standard values and parameterizations proposed in Flake have

been kept. Therefore, an albedo of 0.07 and an extinction coefficient of $1 \ m^{-1}$ was set for water. The fetch parameter which appeared less influent in the sensitivity analysis performed by Bernus et al. (2021), has been parameterized according to the surface area of the lake fraction within the grid cell. Considering a circular pattern for the lake shape, the fetch value has been set equal to the diameter of the circle.

### 3.4 Performed experiments

ORCHIDEE-FLake was run at global scale forced by the various atmospheric datasets presented in Section 2.1.5 and for two different configurations of lake tiling: the first one when all the lakes are grouped in a single class and the second one, where lakes are grouped in three classes according to lake depth. A control simulation corresponding to the standard ORCHIDEE configuration without lakes and using only one atmospheric forcing (CRUJRA) was also performed. Table 1 presents the list of all the simulations performed, the atmospheric forcing used and its spatial resolution, the number of lake categories

and the way the depth was set for each class (median depth value of all the lakes within the tile, arithmetic, quadratic or harmonic averages). All the simulations were run on the period 2000-2016 at a time step of 30 min. A spin up procedure of 20 years on the previous years was applied to bring the prognostic variables to equilibrium. The results were analyzed over the period (2000 – 2016). The name of each simulation is composed of three letters referring to the atmospheric forcing, one number corresponding to the number of lake tiles and four characters referring to the way the depth has been averaged.

Then, the comparison of CJR3med to CJR0 allows to assess the contribution of lakes on the ORCHIDEE surface variables. Comparison of CJR3med to CJR1med allows to see the impacts of the number of lake tiles and the way of averaging is seen in the comparison of CJR1med to CJR1mean, CJR1qmean and CJR1hmean. Finally, the comparison of CJR3med to WCR3med, CNC3med, CNC3med, E2O3med and ERA3med, allows to highlight the impact of the atmospheric forcing and the related uncertainties. The performance metrics, i.e. the RMSE and its components, used in the evaluation are presented in the Appendix

A.



**Table 1.** List of the ten ORCHIDEE-FLake simulations performed at global scale, differing from the atmospheric forcing used and the number of lakes considered

| ID | Meteorological Forcing (spatial and temporal res.) | Number of lake tiles | Depth calculation method |
|---|---|---|---|
| CJR0 | CRUJRA (0.5°, 6 h) | 0 | No lakes |
| CJR3med | CRUJRA (0.5°, 6 h) | 3 | median |
| WCR3med | WFDEI CRU (0.5°, 3 h) | 3 | median |
| CNC3med | CRU-NCEP (0.5°, 6 h) | 3 | median |
| E2O3med | E2OFD (0.25°, 3 h) | 3 | median |
| ERA3med | ERA5 (31km, 1h) | 3 | median |
| CJR1med | CRUJRA (0.5°, 6 h) | 1 | median |
| CJR1mean | CRUJRA (0.5°, 6 h) | 1 | arithmetic mean |
| CJR1qmean | CRUJRA (0.5°, 6 h) | 1 | quadratic mean |
| CJR1hmean | CRUJRA (0.5°, 6 h) | 1 | harmonic median |

## 4 Results

### 4.1 Evaluation at global scale

#### 4.1.1 LWST

The simulated LWST for CJR3med have been compared to the observations provided by GloboLakes for the the 979 lakes matching the ORCHIDEE simulated lake tiles. We recall that CJR3med corresponds to the simulation performed with the meteorological forcing CRUJRA and prescribing three lake tiles differing by their depth (shallow, intermediate and deep lakes). The RMSE calculated on the 17-year period (2000-2016) are plotted in Figure 3 along with latitudinal and zonal averages. Each dot corresponds to a lake of the GloboLakes dataset. Figure 3 shows that the observations are well distributed globally and sampled a large variety of climates and continents. This sampling is representative of the area lake distribution plotted in Figure 2. The RMSE distribution calculated with GloboLakes is therefore a good representation of the ORCHIDEE-FLake model error. Nevertheless, GloboLakes does not provide LWST for frozen lakes, thus LWSTs below $0°C$ are not used in the RMSE calculations. The results show that most of the RMSEs are ranging between 2 and 4 K. Higher errors can be seen for a few lakes with RMSE values up to 8 K. The latitudinal representation highlights a positive trend of the RMSE, when moving from the temperate to the boreal regions. In the tropics, below $20°N$, some large RMSE peak values are observed, which are all the results of individual lakes presenting large errors which impact greatly the zonal average. Because of the low number of lakes at these latitudes, a lake badly simulated has a larger impact on the performance metrics than in the higher latitudes.



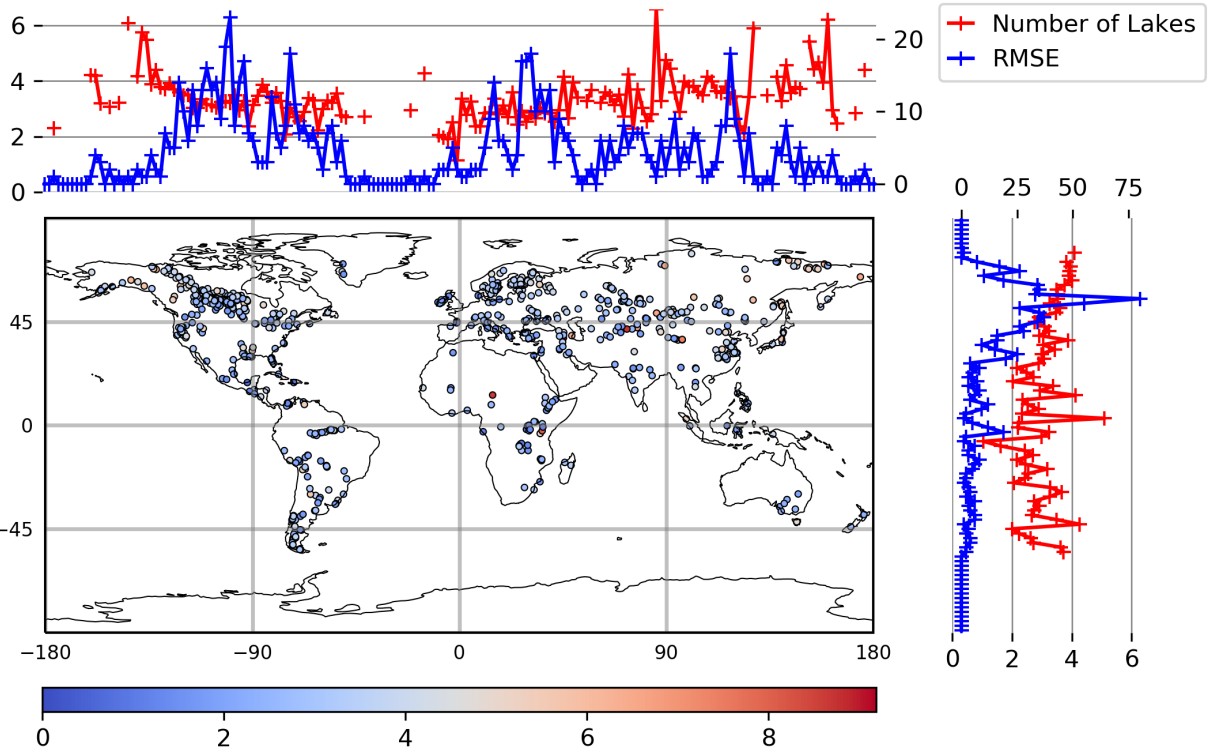

**Figure 3.** LWST errors (RMSEs) simulated by ORCHIDEE-FLake in the CJR3med configuration run at $0.5°$ resolution. Longitudinal(top) and latitudinal (right) averages of the RMSEs (in red) and number of lakes observed in GloboLakes (in blue) are plotted along the map.

### 4.1.2 Ice phenology

Freezing processes were evaluated against the data provided by the Global Lake and River Ice Phenology database over the lakes observed in the northern latitudes. In Table 2, the mean, quartiles and median of the differences between simulated and observed variables are given. The evaluation is shown for three output variables of the CJR3med simulation, which are the start and end of the freezing period (SOF and EOF respectively), and its duration, all in days unit. The results show that the SOF errors are all negative and the EOF errors are all positive. That means that the lakes are generally freezing earlier and melting later in the model compared to the observations. Therefore, the simulated ice duration is larger. The mean bias on SOF is about 11 days (median value), 21 days for EOF, leading to an ice duration overestimated by 32 days in average. The RMSE values are around 24 days for both SOF and EOF and reach 40 days for the ice duration error. The interquartile range is larger for SOF (19 days) compared to EOF (10 days) highlighting the fact that the SOF errors are more scattered. This explains also why the differences between the median and mean values are larger for SOF compared to EOF.





**Table 2.** Statistical metrics (simulations - observations) of the lake ice phenology (CJR3med configuration)

| Statistical metrics | Start of Freezing (days) | End of Freezing (days) | Ice duration (days) |
|---|---|---|---|
| Median | -11.46 | +21.5 | +32.35 |
| Q1 | -4.9 | +26.7 | +43.5 |
| Q3 | -23.9 | +16.3 | +24.1 |
| Mean | -16.46 | +22.1 | +36.0 |
| RMSE | 24.1 | 24.0 | 40.5 |

## 4.2 Sensitivity to meteorological forcing

### 4.2.1 Analysis on specific lakes

Model evaluation has been performed at the local scale for various lakes sampling a large variety of climates and lake properties, in particular their mean depth. Figure 4 presents the comparison of simulated LWSTs to observations for a selection of height lakes. Among those, Victoria and Titicaca are located in tropical regions, Erie, Superior and Baikal in boreal ones, Constance, Tahoe and Tai in more temperate ones. The lake Tai is a shallow lake, the others being deeper (mean depth of each lake is noted on the respective plots). The LWST time series (simulated and observed over the Year 2012) are plotted in Figure 4 along with

the RMSE errors decomposed into its 3 components (Kobayashi and Salam, 2000), i.e., squared bias (SB), squared difference between standard deviations (SDSD) and lack of correlation weighted by the standard deviation (LCS), all represented in bar plots. We recall that the metrics are calculated only for positive values of LWST, since GloboLakes observations do not consider frozen conditions.

Figure 4 shows that the seasonal variations of LWST is generally well represented for all the lakes. The RMSE values are

ranging between 2 and 4 K but can show large variations according to the atmospheric forcing. The dispersion is the largest for the two tropical lakes (Victoria and Titicaca) for which LWST biases can reach 6 K. Biases explain a large part of the RMSE error for the other lakes, especially for Erie and Superior lakes. These results show that the model errors are either explained by biases or by deficiencies to simulate the fluctuation patterns, and less by failure to model the amplitude of the seasonal cycle.

The LWST errors of the tropical lakes are principally explained by biases compared to the other climates because of the

low seasonal temperature amplitude in these regions. The values of the RMSEs range between 1 and 4 K among the lakes. For a given lake, large variations of the RMSEs are observed between the 5 different forcings. For example, the RMSE of the lake Victoria varies between 1 to 4 K depending on the meteorological forcing. The bias SB is the component which could vary the most between each time series (up to 4 K) for the tropical lakes (Victoria and Titicaca). The second component which is significant is the standard deviation LCS, which could reach 1 K (particularly for the lakes under cold climates). The last

component SDSD which allows to diagnose how the model reproduces the LWST amplitude, is always small, less than $0.5K$.




For the temperate lakes, the annual surface temperature amplitude is larger than for the tropical ones, between 15 and 20 K. Tai lake is well represented with RMSEs below 2 K for most of the forcings. For the two others (Tahoe and Constance), the model fails to simulate the annual amplitude which is underestimated and the cooling at the end of summer is delayed whatever the forcing used. Such errors could result from parameterization errors, such as depth errors which are known to

influence greatly the seasonal LWST pattern. Even if the freezing periods are not accounted in the RMSE calculations, the time series highlight the large impact of the atmospheric forcing on the timing and amplitude of the ice cover. For example, if we consider the Erie lake which did not freeze in the observations in year 2012, the winter temperatures are very well modeled with all the forcings except when CRUNCEP is used. This failure has been explained by the cold bias of the air temperature identified in CRUNCEP over this part of Canada when compared to the other forcing datasets (not shown here).

### 4.2.2    Global scale analysis

Figure 5 presents the performance metrics calculated on all the lakes observed in GloboLakes that were represented in OR-CHIDEE in the correct depth class tile (i.e., 979 lakes all other the world). The statistics were calculated for the seventeen years of the whole simulation period (2000-2016). The metrics shown here are the RMSE, the unbiased RMSE, and the three components of the total RMSE, i.e., SB, SDSD and LCS. The values obtained for each lake and for each simulation are com-

pared to the ensemble mean in the scatterplots. Therefore, these plots inform on the sensibility of the surface temperature to the atmospheric forcing. In addition,the boxplots show the metrics calculated for each simulation forced by the five atmospheric forcings used.

The results show that the simulations forced by the meteorological reanalysis with the lower resolution (i.e., $0.5°$) lead to the worse performances (higher RMSEs (between 3.18 K for CRUJRA, 3.16 for CRUNCEP and 2.82 for WFDEI-CRU). When

higher resolution forcings are used (E2OFD and ERA5), the RMSEs decrease to 2.81 K and 2.69 K. An important part of the errors is therefore explained by the resolution and accuracy of the meteorological forcings. Thus, the model errors will be more complicated to evaluate.

However, the decomposition of the RMSE is quite similar among the forcings on average. The scatterplots on the left panel compare the mean RMSE for the different forcings. The high dispersion on the scatterplot shows that the modeled surface

temperature is sensitive to the uncertainty of the atmospheric forcing. As a matter of fact, the LCS and SB components explain about 50% of the RMSE whatever the forcing used, the SDSD component being minor. More precisely, LCS and SB explain at least 50% of the RMSE for more than 50% of the observed lakes when SDSD explains less than 20% of the RMSE for more than 75% of the observed lakes. Thus, the LWST errors are firstly explained by the model biases and deficiencies to reproduce the seasonal patterns, confirming what was already noted for the individual lakes analyzed previously. In Figure 5, the signed

bias is plotted in the scatter plot whereas, in the whisker plot the squared bias (SB) normalized as the other components is plotted. Then, it is clear that for all the forcings, the bias is mostly positive, meaning that the model generally overestimates the surface temperature. Compared to the total RMSE that can reach up to 10 K, the unbiased RMSEs are reduced and do not exceed 6 K. The dispersion is also less important than the one observed with the standard RMSE. Indeed, the standard



**Figure 4.** Time series of the daily mean lake surface temperature simulated for eight lakes with five different meteorological forcings compared to GloboLakes observations for the year 2012. The barplots represent the decomposition of the RMSEs in terms of bias, lack of correlation and difference of standard deviations between model and observations.





**Figure 5.** LWST errors obtained on the 979 lakes observed in GloboLakes for the five simulations CRJRA3med, WCR3med, CNC3med, E2O3med and ERA3med. RMSE, unbiased RMSE, LCS, SDSD and sign SB are represented as scatterplots (left panel) in comparison to the ensemble mean and in boxplots (right panel) where the median is represented by the orange line, and the quartiles by the edge of the color box





deviation of the values to the identity line ($\sqrt{\frac{1}{N}\sum(RMSE_i - RMSE_{Mean})^2}$) is 0.47 on average for the RMSEs and 0.20 on
average for the unbiased RMSEs. Therefore, the model biases explained a large part of the LWST errors.

The sensitivity of the simulated ice phenology (SOF, EOF and ice duration) to the meteorological forcing has been evaluated
and the results are plotted in Figure 6. The whisker plots show that the impact of the meteorological forcing is lower on the
prediction of the start of lake freezing (SOF) than on the prediction of the end of the freezing period (EOF), and consequently on
the ice duration where the two errors cumulate. It can be seen also that the errors are smaller for the two simulations performed
with the higher resolution forcings (E2OFD and ERA5) with a significant advantage of E2OFD especially for the simulation of
EOF. The SOF, EOF and duration median errors are respectively equal to -6, 11 and 20 days with E2OFD whereas they were
up to -14, 25 and 41 days with the CRUNCEP forcing.

### 4.3 Sensitivity to lake depth parameterization

To understand the impact of the depth parametrization, we analyse the differences between the real depth of the observed
lakes and the depth prescribed in the corresponding ORCHIDEE tile at a grid resolution of $0.5°$. A preliminary step to assess
this error is to assign the lake from HydroLAKES to the lake from GloboLakes. We succeed to link 964 of the 979 lakes
of GloboLakes. The fifteen missing lakes which were not present in HydroLAKES have not been included in the evaluation
procedure. When GloboLakes and HydroLAKES are in agreement over a single lake, the depth referenced in HydroLAKES
was assigned as the observed depth. But when several water bodies in HydroLAKES correspond to a single observed lake,
we assign the area weighed mean depth to assess the depth of the observed lake. Figure 7 presents the distribution of the
errors between the observed depth as defined previously and the depth prescribed in the ORCHIDEE tile for the five model
configurations, i.e., depth median value calculated in a 3-lake tile configuration (3med) and in the 1-lake tile configurations
(1med, 1mean, 1hmean and 1qmean). The probability density functions (PDF) and the cumulative distribution functions (CDF)
are both plotted.

The configuration with 3 tiles shows the lowest errors on the depth parameterization, and the median appears to be the best
way of depth aggregation within a grid cell: about 90% of the observed lakes have an absolute error less than 2.2 m when the
depth parameter is set to the median value within the lake tile. In comparison, 90% of the observed lakes have an absolute error
less than 4,4 m the quadratic mean and than 4.9 m with mean depth estimation approach and the harmonic mean is the worst
approach since only 90% of the observed lakes have an absolute error less than 10.2 m.

In Figure 8, the total RMSEs, the unbiased RMSEs, SDSD, LCS and the signed bias are plotted as scatterplots against
the ensemble mean and as box plots. Compared to the previous sensitivity analysis to the atmospheric forcing, the errors are
slightly lower as well as the dispersion among the simulations. The RMSE values are generally lower than 6 K with a median
value between 3.1 K and 3.2 K. The components present similar order of magnitude among the simulations and the scatterplots
show a low dispersion around the identity line for each component. This result was expected since the impact of the lake depth
parameter in FLake, impacts mostly the pattern of the seasonal cycle of LWST and to a lesser extent, its amplitude.

The ice phenology sensitivity to the number of lake tiles and depth parameterization strategy is presented in Figure 9. Here
also, the results show less impact compared to the sensitivity obtained with the meteorological forcing. In contrast, it appears



**Figure 6.** Boxplot of the RMSE errors on the start, end and duration of the ice period for the five simulations CJR3med, WCR3med, CNC3med, E2O3med and ERA3med.



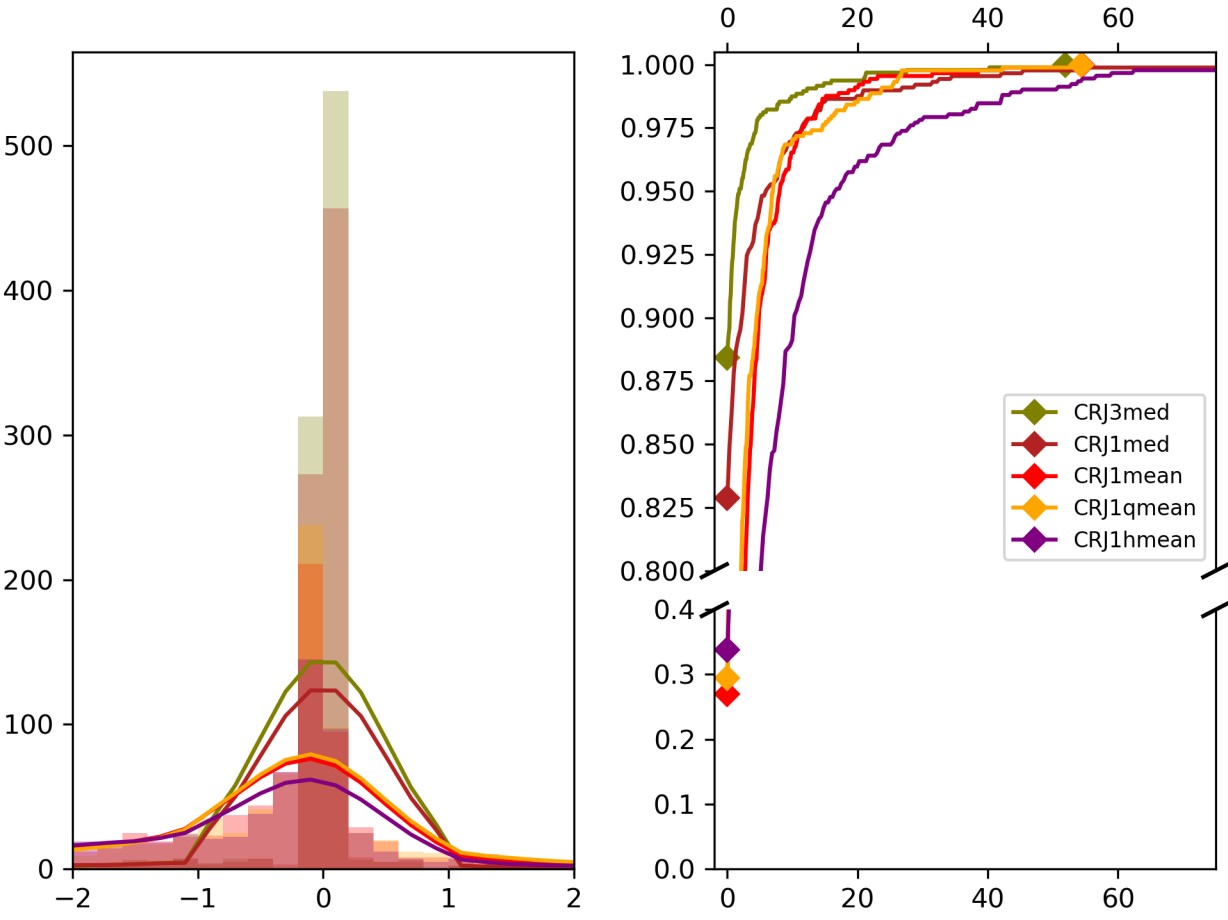

**Figure 7.** The probability density function (left panel) and cumulative density function (right panel) of the depth errors between the observed and simulated lake for each depth parameterization strategy.

that the SOF errors are reduced by half in the 1-tile configurations compared to the 3-tile one. In the four 1-tile simulations, SOF errors are around 5 days whereas it is equal to 10 days in the CJR3med simulation. However, the variance is quite

similar, the 3-tile simulation shows a clear bias not observed in the 1-tile simulations. EOF and ice duration errors are similar among the five simulations, equal to about 22 days and 27 days respectively, comparable to what is obtained in the CJR3med simulation. The improvement observed in the simulation of SOF in the 1-tile simulations compared to the 3-tile one, despite the increased depth errors, can be explained by the model biases. As a matter of fact, we highlighted in Section 4.2.1 the systematic underestimation of SOF time: LWST cooling at the end of summer is too rapid and the lake freezing is earlier compared to

the observations. If we consider that the shallow lakes represent a larger fraction of the lakes which are most likely to freeze and that the depth parameter of this class is the most impacted in the 1-tile configuration, it is clear that parameter errors are



**Figure 8.** LWST errors obtained on the 979 lakes observed in GloboLakes for the five simulations CJR3med, CJR1med, CJR1mean, CJR1qmean and CJR1hmean. RMSE, unbiased RMSE, LCS, SDSD and SB are represented as scatterplots (left panel) in comparison to the ensemble mean and in boxplots (right panel)





compensating model deficiencies. As a matter of fact, depth of shallow lakes, is most likely to be overestimated in the 1-tile configuration. The deeper the lake, the later the start of freezing. Therefore, the depth overestimation will help to compensate the too early cooling. The impact of this error is less visible on the EOF because the depth parameter is less influential during
the ice melting process (Bernus et al., 2021).

## 4.4  Energy balance impacts of lakes in ORCHIDEE

To analyse the impact of representing lakes on the global energy and water budgets, the two simulations CJR0 et CJR3med were compared at the global scale. The differences (CJR3med-CJR0) in terms of surface temperature, latent and heat fluxes averaged over the grid cell, were calculated for the four seasons of the year and plotted in Figure 10. Winter period is defined
as the average of the December, January and February months, summer as the average of June, July and August.

The comparison of the two simulations (with and without lakes) show interesting features. First, the surface temperatures appear warmer during a large part of the year, except for summer where the regions presenting large fractions of lakes of intermediate depth, appear colder. In the high latitudes during the winter and autumn seasons, the temperature is generally warmer for all kind of lakes. For the deeper lakes, this difference can reach 10 K. It is the case for the Baïkal lake whereas for
the Michigan and Superior lakes, the difference is between 5 and 7 K. In the regions covered by smaller and shallower lakes in Scandinavia and Canada, this difference remains below 2 K due to the lower lake fraction and smaller smoothing effect of shallower lakes. In spring, the deep lakes as the great American lakes or the Baïkal lake are cooler than their environment and explain the differences of temperature between the two simulations by 5-6 K and 3-4 K respectively. However, the regions where the lakes are shallower remain warmer in the lake simulation because of their lower thermal inertia (but only by 0.5
K). In summer, most of the lakes are cooler in the northern latitudes compared to a neighbouring bare soil and the differences between the two simulations are always positive. In tropical latitudes, the pattern is different. During the humid season between October and April, the simulation with lakes is warmer at the end of the humid season by 1 or 2 K (in Africa during the spring season) and cooler during the end of the dry season by 0 or 3 K (in Africa).

Averaged at yearly time scale, the simulated global evaporation is 1.07 mm/d with lakes and 0.82 mm/d without lakes.
Focusing on the summer period, the global evaporation rises up to 1.72 mm/d with lakes and to 1.50 mm/d without lakes when lakes occupy at least $20\%$ of the cell fraction. Then, the contribution of lakes rises the summer evaporation by 0.22 mm/d. For a grid cell completely covered by water, i.e., lake fraction is equal to 1, the evaporation difference can reach 0.31 mm/d. In tropical areas, during the summer period and for latitudes below $30°$, this difference is larger and equal to 1.16 mm/d when the lake fraction is above $20\%$ and amount to 1.6 mm/d when the lake fraction is equal to 1.
In terms of fluxes, the latent heat flux is generally stronger everywhere during part of spring and summer as it is showed by the rise of the mean evaporation rates. In spring and in the northern latitudes, the latent heat flux is lower in the simulation with lakes, because of the freezing processes which prevent the lake evaporation. Compared to a bare soil that start to evaporate earlier in ORCHIDEE, the lake evaporation is lower in early spring. For example, the latent heat flux of Baikal lake is between 25 and $50\,W/m^2$ less than the bare soil evaporation and it is about the same for the Superior lake (between 20 and $40\,W/m^2$).





**Figure 9.** Boxplot of the RMSE errors on the start, end and duration of the ice period for the five simulations CJR3med, CJR1med, CJR1mean, CJR1qmean and CJR1hmean

**Figure 10.** Differences (CJR3med - CJR0) in the surface temperature (first line) latent heat Flux (second line) and sensible heat flux, simulated with and without lakes, averaged over 3-month periods starting from December (one per column)





The freezing duration is longer for a lake than for a bare soil. The sensible heat fluxes are generally weaker everywhere during part of spring and summer because of the larger latent heat fluxes in the simulation with lakes.

## 5   Discussion and conclusion

This work presents how lakes energy balance is now represented in the ORCHIDEE land surface model, thanks to the FLake lake model. The methodology and the model evaluation at global scale were presented. Several simulations were performed

to evaluate the model sensitivity to the meteorological forcing and to the lake tile depth parameterization. The evaluation of LWST and ice phenology against the GloboLakes dataset and the Global Lake and River Ice Phenology Database, showed that the lake surface temperature is quite well simulated in this new ORCHIDEE version. Compared to the observations, LWST errors range between 2 and 4 K, even if in some few cases, the errors can reach 8 K. These results are comparable to what was obtained by previous works, such as Dutra et al. (2010) who obtained the lake RMSE median around 3.75 K in the high

latitudes ($> 40°$) and around 2.25 K for low latitudes ($< 40°$). The lake ice phenology evaluation showed that the ice/snow coupled processes are also fairly well simulated with ice duration errors explained partly by start of freezing errors. Here also, compared to previous works, the results obtained are quite similar. As an example, Dutra et al. (2010) obtained a SOF error of 30 days (in advance) for $75\%$ of lakes after optimisation of the depth parameter against LWST observations. The expected impacts of lakes on the near-surface atmospheric variables have been assessed by comparing the grid cell averaged

surface temperatures and fluxes simulated by ORCHIDEE, considering lakes energy balance or not (in that case, lakes were considered as bare soils). The results show larger evaporation rates at the annual scale (around 0.25 $mm/d$ ) and smoother LWST seasonal amplitudes. The impacts of these added fluxes on the atmosphere should be significant at regional scales, in particular on precipitation predicted fields. As an example, Van de Walle et al. (2020) showed that Victoria Lake evaporation explains up to $50\%$ local precipitation and storm events. The addition of lakes in the IPSL climate model is expected to produce

the same impacts, helping to better reproduce the near-surface variables, the precipitation regimes and the extreme events.

### 5.1   Atmospheric forcing uncertainties

A large part of the LWST errors is explained by the atmospheric forcing uncertainties. We have examined in more detail, the lakes which present the largest RMSEs. Among the 969 lakes studied, only 15 present large RMSEs, larger than 6 K. For those lakes, we observed strong biases in the atmospheric temperatures (whatever the reanalysis product) which explain the

largest part of the error. If we consider the whole dataset, our results show the large sensitivity of LWST to the atmospheric conditions and the large uncertainties of the reanalysis products. When looking to the RMSE decomposition, biases were explaining more than $50\%$ of the error for more than $50\%$ of lakes. It was particularly the case for the tropical lakes where the LWST seasonal amplitude is lower than for higher latitudes conditions. The model sensitivity to the atmospheric conditions is also very strong during the intermediate seasons for the lakes which are subject to freezing. Because of the non-linearity of

these processes, a small bias on the air temperature can completely change the temperature evolution of the lake, and greatly influence the ice phenology. The large sensitivity of the SOF/EOF times to the atmospheric forcing shown in our simulations,




is a good demonstration of this effect. Therefore, it appears very necessary to use various atmospheric forcing products in the model evaluation, since they all contain biases, which on top of that, present regional features. These biases are the result of the different datasets used in the generation process, which consists in merging model simulations and multiscale observations
through data assimilation technics. It should be noted that the land surface models used in this process do not account for lakes and are constrained by static land cover maps which are not updated regularly. Biases are also linked to the spatial and time resolution of the datasets. It has been shown that LWST errors were smaller when the two best resolution forcings are used (E2OFD and ERA5). Increasing the spatial resolution improved LWST RMSEs by 0.5 K. Grid resolution is indeed an important parameter for the modelling of land surfaces which have very different functioning than the rest of the environment.
Moreover, the atmospheric conditions above a lake surface are different than in the surroundings, complicating greatly the model evaluation at global scale. The use of various reanalysis products is therefore the only way to separate forcing and model errors.

**5.2 Model parameterization**

Systematic model errors appeared among the simulations and could be the result of parameterization errors. Sensitivity to depth
parameterization was highlighted in previous works. Our simulations did not show a large impact of the depth parameterization at global scale on the LWST error statistics, but the impacts on the ice phenology was shown to be significant, especially in the determination of the time of the start of freezing (SOF). As already said, lake depth is determinant on the lake freezing process but less on the melting one which are more driven by the atmospheric conditions and the amount of snow than directly on the lake features.
Scaling issues may explain also part of the errors : FLake is a 1-D model and therefore, do not consider the spatial variability of the within-lake thermal processes and properties, nor the temporal variability of the parameters. In particular, the lake depth is a constant parameter in FLake whereas it is clear that in reality, it varies spatially and temporally, and so LWST and ice phenology. Lakes subject to anthropic management as reservoirs see their surface and water level changing throughout the year. The averaged depth and surface area referenced in the databases do not report these changes. For example, the lake Chad
in HydroLAKES has a surface area doubled compared to the current value and the average depth is probably erroneous as well. If we consider the ice observations, SOF time corresponds to the start of freezing in the observations whereas it can take some days before the lake is completely frozen. In contrast in FLake, because of the 1-D assumption, the lake surface is totally frozen in a time step and so abruptly will vary the related radiative parameters. For big lakes with large surface areas, this scaling error may be important. Similar conclusions were drawn by Pietikäinen et al. (2018) who identified two factors explaining the
delay in their simulated EOF and the advance of SOF with the coupled FLake-REMO model. Firstly, the lakes are freezing too early and the snow accumulation can start earlier. An overestimation of the snow cover may explain the delay in the timing of the end of freezing and the overestimation of the ice duration period. The prescription of a constant average value of the snow conductivity in our simulations, could also play a role in the deficient representation of the dynamics of the ice/snow melting. Secondly, processes as the snow erosion and transport by the wind are not modelled and may lead to an overestimation of the
snow accumulation in the simulations.





Scaling and parameterization issues may affect other parameters like the radiative ones : albedo and extinction coefficient which were both prescribed as constant values in space and time. This assumption may impact also the model performances. Recent databases can be used to solve this issue, based on satellite measurements. Albedo products could be used to spatialize the albedo parameters. The extinction coefficients are also known to be key (Heiskanen et al., 2015). For example, Zolfaghari

et al. (2017) demonstrated the improvements brought by constraining the extinction coefficient of the Superior lake using FLake, based on remote sensing observations.

Missing processes could also explain part of the errors such as the impact of water salinity, which could present seasonal and interannual variations linked to human activities and water level changes. In our evaluation, among the fifteen lakes presenting the largest errors, three of them are saline lakes (Issyk-Kul, Urmia and Natron) with high salinity levels and which can present

salt crust regularly ( Zavialov et al. (2018), Sima et al. (2013), Tebbs et al. (2013)). Salinity impacts evaporation (which is lower compared to those of freshwater), and consequently the surface temperature and its seasonal evolution. As reported by Ahmadzadeh Kokya et al. (2011), the temperature rising during the summer can be more rapid for saline lakes, as they observe for the Urmia lake in Azerbaijan. The consideration of salinity in the water freezing point could improve the results, especially the cryospheric processes.

## 5.3 Future work

Our results have shown the potential of ORCHIDEE-FLake coupled model to represent the energy balance of lakes, the surface temperature and fluxes at the atmosphere interface. The model performances are satisfactory and the various sources of errors are well identified. The flexibility of our developments allowing to prescribe a variable number of lake tiles within a model grid cell, open new perspectives for improvement. First, we intend to study the spatial and temporal variability of the lake radiative

parameters and to implement them in the model. Data from high spatial resolution optical sensors such as Sentinel-2 with a 10-m resolution are very promising and some products have recently came out such as Wang et al. (2020) on the lakes of the Tibetan plateau, or Zolfaghari et al. (2017) on the Erie lake.

Lake area and depth are key parameters of the model. Thanks to satellite altimetry, water level changes as well as surface areas can be monitored. Future instruments as the next SWOT mission will soon provide these data at the global scale for lakes

larger than 250 m x 250 m with a temporal resolution varying along the swath but better than 21 days (Biancamaria et al., 2016). Such data will be of tremendous interest to improve the calibration of both lake depth and area in FLake. Besides, the data will be also key to constrain water volumes, since resolving the lakes mass budget is the next development planned in our model.

Our preliminary results have shown that lakes are expected to impact the near-surface atmospheric fields. These impacts

have already been shown by numerous studies such as the one of Le Moigne et al. (2016). The coupling of ORCHIDEE-FLake to the atmospheric model LMDZ is another mid-term perspective that will require developments in LMDZ to account for different atmospheric columns within an ORCHIDEE grid cell. But before, simulating the thermal processes in lakes provides the possibility to model other biogeochemical cycles, in particular GHG fluxes like carbon and methane. Then, the




tools developed are opening the path to studies on the contribution of lakes to global warming and impacts of climate and
environmental changes.

*Code availability.* Code for this project is available at https://forge.ipsl.jussieu.fr/orchidee/wiki/GroupActivities/CodeAvalaibilityPublication/
ORCHIDEE_FLAKE_gmd_2021.

**Appendix A: Performance metrics**

Bennett et al. (2013) describes several methods to characterize model performances. A simple way is to use an indice such as
the Root Mean Square Error (RMSE) which is widely used to measure the performance of a model and is defined by Equation
A1.

$$RMSE = \sqrt{\frac{1}{N}\Sigma(T_{sim} - T_{obs})^2} \tag{A1}$$

To better understand the errors, we decomposed the MSE (the square of the RMSE) into three components: the squared bias
(SB), the lack of correlation weighted by the standard deviation (LCS) and the squared difference between standard deviations
(SDSD). The demonstration of this decomposition is done with the method described by Kobayashi and Salam (2000). LCS
evaluates if the model correctly reproduces the temporal pattern and SDSD if the model gives the correct temporal amplitude.
We also add the signed squared bias (SSB)

$$\begin{aligned}
LCS &= 2SD_{sim}SD_{obs}(1-r)\\
SDSD &= (SD_{sim} - SD_{obs})^2\\
SB &= (\overline{T_{sim}} - \overline{T_{obs}})^2\\
SBB &= \text{sign}(\overline{T_{sim}} - \overline{T_{obs}})\,SB
\end{aligned} \tag{A2}$$

with

$$\begin{aligned}
SD_{sim} &= \sqrt{\frac{1}{N}\sum(T_{sim} - \overline{T_{sim}})^2}\\
SD_{obs} &= \sqrt{\frac{1}{N}\sum(T_{obs} - \overline{T_{obs}})^2}\\
r &= \frac{\frac{1}{N}\sum(T_{sim} - \overline{T_{sim}})(T_{obs} - \overline{T_{obs}})}{\sqrt{\frac{1}{N}\sum(T_{sim} - \overline{T_{sim}})^2}\sqrt{\frac{1}{N}\sum(T_{obs} - \overline{T_{obs}})^2}}
\end{aligned} \tag{A3}$$

$$LCS = \frac{\sqrt{\frac{1}{N}\Sigma(\overline{T_{sim}}^F - T_{obs})^2}}{\frac{1}{N_F}\Sigma(\sqrt{\frac{1}{N}\Sigma(T_{sim}^F - \overline{T_{sim}}^F)^2})} \tag{A4}$$





In some studies, the errors between the simulations and the observations are evaluated after removing the bias. This metrics is the unbiased RMSE ($RMSE_{unbiased}$) defined with Equation A5. We also used this metric in our study.

$$RMSE_{unbiased} = \sqrt{\frac{1}{N}\Sigma(T_{sim} - T_{obs} - \text{sign}(\overline{T_{sim}} - \overline{T_{obs}})\sqrt{SB})^2} \qquad (A5)$$

*Author contributions.* AB and CO designed the research, AB made the model developments, CO provides supervision and funding acquisition, AB and CO analyzed the results and wrote the manuscript.

*Competing interests.* The contact author has declared that neither they nor their co-authors have any competing interests.

*Acknowledgements.* This work was supported by the French National Space Agency (Center National d'Etudes Spatiales) through the TOSCA-SPAWET program in preparation to the SWOT space mission. CEA (Commissariat à l'Energie Atomique et aux énergies alter-
natives) is also aknowledged for funding part of A. Bernus PhD grant. The authors would like to thank Karine Pétrus for her preliminary works with FLake model in our team and Fabienne Maignan for her precious advices on coding issues.





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
