# Peer review of "Modeling subgrid lake energy balance in ORCHIDEE terrestrial scheme using the FLake lake model"

_Geoscientific Model Development, 2021_

## Author Response (AR1)

**Reviewer 1:**

Modeling subgrid lake energy balance in ORCHIDEE terrestrial scheme using the FLake lake model by Bernus and Ottlé, 2022.

This article describes the implementation of the FLake lake model in ORCHIDEE. A subgrid approach has been adopted to represent lakes of different depths in the same model mesh. This ORCHIDEE-FLake coupling was evaluated with long offline simulations using reanalyses at resolutions of about 25 to 50 km. Thus, surface temperature and ice phenology have been evaluated on a thousand lakes. It is shown that the variability of atmospheric forcing has a strong impact on the dispersion of the results. Moreover, the tile approach for lakes of different depths has a significant effect on the ice formation date compared to a simulation where this subgrid approach is not activated. The impact of the lake tiling on the surface temperature is not significant.

This article describes a work to improve the representation of the surface and its processes by implementing a lake module. The work is based on work already done in other models and brings a novelty with the subgrid representation of lakes. This work can be tested in the LMDZ climate model and will undoubtedly help to explain or even reduce some of the temperature biases in high latitudes.

The article is well structured and clearly shows the contribution of the subgrid approach adopted to represent the lakes in this multi-tile system. However, some parts need to be clarified and my major comments concern the quality of the figures which is not sufficient for some of them and therefore deserves to be improved for a better reading of the manuscript. Data availability is not described.

Response: We thank you for reviewing our manuscript and sharing your suggestions and comments with us. We have thoroughly revised our paper following your advices and especially re-done and improved the quality of all the figures which are now all in the portrait mode in our revised version.

Major comments:

The figures need to be improved. In particular, the units of the plotted fields are often missing (Fig. 2,3,4,5,7,8,10). For some of them the size is too small for a correct reading.

P11: Figure 2 is not satisfactory. Reading in landscape mode is not comfortable for the reader. A portrait orientation would be better. Also, the legend is wrong. Then add a, b, c, d to each of the panels and refer to them in the legend and in the discussion. The size of the thumbnails makes it almost impossible to read them: in my opinion we can remove the first map which lists all the lakes (depths from 1 to 500m) and be satisfied with the 3 classes, then choose the portrait mode and put 6 maps on the vertical: depth 1, fraction 1, depth 2, fraction 2…

Response: The new figure includes all these modifications. First, we have deleted the first case with all the lakes. Second, we have followed the recommendations on the figure layout. Third, we have corrected the legend.

P14: the figure 3 deserves to be improved. Units are missing on the longitudinal and latitudinal mean graphs. On the top right graph, 0, -10 and -20 appear: why this "-" sign? The legend is reversed (colors).

Response: In the new figure, we added all the label axes and the colorbar label. We have changed with a color more appropriated . We have added a vertical axis to take off the illusion of "-" in front of the 0, 10 and 20.

P19, L405-409: the errors cannot be read easily from figure 7, especially for small errors. A table containing the errors with in x the depths between 0 and 10m every 2m then between 10 and 20m every 5m and finally every 10m up to 50m, and in y the cumulative density function for 95, 90, 85, 80 and 50% the values of the errors as a function, would help in reading figure 7.

Response: The new figure 7 should be more readable with a new color scale and larger characters.  We therefore, don't think that it is necessary to add a table besides.

p25: figure 10 is difficult to read because it is too small, and the maps are presented in landscape mode, which does not facilitate reading. You have to enlarge each thumbnail (you have to zoom in +400% to see something, and some readers still use the paper format to read the articles). Also, the units are missing on the color bars.

Response: We have added a title on the colorbar and have enlarged each thumbnail. The figure is now in portrait mode and the missing units have been added.

The data availability is mandatory in many journals and GMD attaches great importance to this aspect. In this manuscript, this section is missing and should be added.

Response: This section has been added in the revised version of the manuscript. All the data used are on open access and available online. The data of our simulations are available on request to the coauthors.

Minor comments:

P3, L60: MacKay's model did not participate in the LakeMIP intercomparison exercise.

Response: Yes, you are right,  it is a mistake that was corrected in the revised manuscript.

P3, L74: MLake was not developed in SURFEX but in the SURFEX environment to allow for water exchange between lakes, rivers and land surfaces.

Response: Corrected.

P5, L127: the different forcing are presented and later it is said that their variability is large and that they have a strong impact in particular on the surface temperature. To clarify this point it would be good to show how the forcing differ: for example, by showing average annual cycles of air temperature, radiation…

Response: It is difficult to map the variability of all the forcings at the global scale, since the differences/biases vary in time (seasonally) and spatially for all the variables. We have seen

that depending on the lakes and on the climate encountered, the differences can be explained mostly by the air temperature or the longwave radiation. We have added this information in the discussion of the results, where the impact of the atmospheric forcing variables was already discussed .

P5, L140: the methodology is not explicitly presented under this link. Please give more details on the methodology.

Response: This sentence was misleading and it is better to remove the link, the methodology is described in Viovy (2018) and we added another reference (Wei et al., 2014) which better describes the methodology : Wei, Y., Liu, S., Huntzinger, D. N., Michalak, A. M., Viovy, N., Post, W. M., Schwalm, C. R., Schaefer, K., Jacobson, A. R., Lu, C., Tian, H., Ricciuto, D. M., Cook, R. B., Mao, J., and Shi, X.: The North American Carbon Program Multi-scale Synthesis and Terrestrial Model Intercomparison Project – Part 2: Environmental driver data, Geosci. Model Dev., 7, 2875–2893, https://doi.org/10.5194/gmd-7-2875-2014, 2014.

P6, L163: missing precipitation forcing that play a primary role when the snow module is activated.

Response: Sorry, precipitation is indeed playing a key role in freezing conditions when snow can accumulate. It was forgotten in the list of the forcing variables and has been added in our revised document.

P7, L178: the snow depth is calculated through an evolution equation that considers the snow precipitation rate in the time step.

Response: Sentence corrected.

P9, L236: "time split": how is this effect implemented in the coupled model? Is it in FLake, in the ORCHIDEE driver, or in a call interface to FLake? Please give details on this aspect, maybe proposing an appendix that describes it if it is justified.

Response: The time split was implemented in FLake (only on the snow module) and not in the ORCHIDEE driver with a simple loop. We added it in the revised version.

P10, L255: the permanent water surfaces come from ESA-CCI? If so, please specify.

Response: The permanent water surfaces of the CCI-Land Cover database has been removed from the ORCHIDEE PFT maps. The lake fraction in the grid cell was derived from the HydroLAKES database which provides the depth of the water bodies at the same time. In order to have the sum of the various land cover fractions equal exactly to unity, we have imposed the HydroLAKES fractions and rescale the other grid fractions coming from CCI. We have reformulated this sentence in order to be clearer.

P12, L276: The Caspian Sea is treated as a lake: what motivated this choice?

Response: In the ORCHIDEE land surface model when coupled to the other modules of the Earth System Model, the Caspian sea is simulated thanks to the ocean model NEMO. When ORCHIDEE is run offline, we have chosen to apply the FLake model, which is better than to

consider that the Caspian sea is a bare soil, like it is done in the previous version of the code. We can note that in some ESMs, the Caspian sea is simulated with a lake model instead of an ocean model and has shown to give better results (see paper of Choulga et al., 2019  fo example )

P15, A329: the comparisons presented in this section refer to the year 2012 with no reason given. How does 2012 compare to other years? Do we see the same annual cycles? How do they differ? The results would have been more robust if Figure 4 had presented the results of the average annual cycle calculated over 2000-2016 as for 4.2.

Response: In figure 4, we present the comparison of LWST simulations to observations. Because of the interannual variability of the forcing data and of the observations, a mean average seasonal cycle over a number of years is certainly  interesting to highlight systematic biases but less to discuss the model-data discrepancies given the observations uncertainties (better assessed by looking at daily variability). It is difficult to plot more than one year, and we chose 2012 as an example. The biases that we have highlighted are present in other years. In the revised version, we have finally switched to 2010, because there were more data available on lake Victoria, but the conclusions were pretty the same as you will see.

P15, 341: there are no continuous observations on Lake VIctoria: any explanation? Is this the case for other years between 2000 and 2016 and how does it affect the statistical results?

Response: It was the case for Year 2012 but not for all the years. So, we have switched to 2010 presenting more observations in the revised figure to show longer time series.

P16, L359: it would be interesting to add a comment on Baikal which is not frozen until mid-December while the model has a very cold temperature and therefore a strong negative bias.

Response: The comment was added in the revised version. The differences are probably linked to the limits of FLake 1D freezing model which do not account for fractional ice cover. This point is also more deeply discussed in the discussion part of the revised manuscript.

P16, L364: the decomposition of the RMSE into SB, SDSD and LCS should be better highlighted in the appendix. What is the relationship between the 3 and how to interpret them should be added?

Response: The sum of the 3 components is equal to the MSE, we have modified in the appendix the presentation of the decomposition and hope that it is more clear in the revised version.

P16, L365: comparison to the ensemble mean is not clear: what I understand is that each point is the RMSE (e.g.) calculated for each forcing for the 1000 selected lakes. But the ensemble mean is the average of the RMSEs of the 5 simulations, right? It should be clarified in the text. The units on the y-axis should be added.

Response: We have better explained what is plotted in this figure in the revised version. The figure was redone and the units were added

P16, A375-378: the explanation is not clear: what shows that LCS and SB explain 50% of the bias? The reader should be guided.

Response: This part was rewritten to better explain what is plotted (see lines 400-403).

P16, L381: "the bias is mostly positive": this conclusion is not obvious when reading figure 5: the signed bias varies from -1 to +1 and on the right it is $SB^2$ which is plotted, thus positive.

Response: The figure has been revised to better show the differences between the simulations. We hope that it is more clear in the revised version

P19, L390: Why does the E2OFD forcing have better scores than the others? Which component of the forcing plays the most?

Response: We can not give a clear answer, we guess that the better scores obtained by the 2 best resolution forcings are the result of their higher spatial resolution which allows to represent the atmospheric conditions above the lake surfaces in a more realistic way. But since we do not have any in situ atmospheric data to evaluate the forcings, it is difficult to be more affirmative.

P19, L403: it is more a histogram than a PDF for which the sum of the values is 1.

Response: Agree, it has been corrected in the manuscript.

P23, L432: CRJ and CRJ3med do not analyze water budgets, but rather surface energy balances (latent heat flux is an energy flux).

Response: Sorry, the sentence was misleading and was corrected in the revised version.

P27, L495: "used in this process": to be reworded to clarify the point

Response: Will be corrected

P27, L512: and so do LWST and ice phenology

Response: Agree and corrected

P29: add the link between RMSE and the other components discussed in the manuscript; the superscript "F" appears in the LCS formulation and has not been defined

Response: Corrected

Technical comments:

P3, L62: models instead of Models

Response: The change has been done

P5, L118: the ARC-Lake database

"project" was changed into "database"

P5, L120: extra parenthesis after 0.7 K; bias instead of biase; "for day" and "for night" could be replaced by "during daytime" and "at night"

Response: The parenthesis has been taken off,  the "e" has been removed and for day and for night has been replaced

P5, L132: a space is missing before WFDEI

Response: The space has been added

P7, L185: units should be homogenized throughout the manuscript (degrees Kelvin and not Celsius, superscript notation, italics...)

Response: Done

P10, L263: use etc. or ... but not etc…

Response: "etc" has been removed

P12, L279: FLake instead of Flake

Response: "l" has been replaced by "L"

P12, L298: CNC3med appears twice.

Response: The second occurrence has been deleted

P14, A324: put a "-" sign in front of 11 days

Response: the sign has been added

P15, L331: eight instead of height

Response: "h" has been removed

P15, L334: year instead of Year

Response: The capital letter has been removed

P16, L362: corresponding instead of correct?

Response: "corresponding" is more suitable, so it has been changed

P16, A369: extra parenthesis before between; 3.16 and 2.82 shall be followed by K

Response: The parenthesis has been taken off and the units have been added

P19, L384: missing units for RMSE

Response: The units have been added

P19, L402: median depth

Response: "depth" and "median" have been swapped

P23, L453: 30°N

Response: "N" has been added

P23, L459 Lake Superior

Response: It has been changed

P27, L495: techniques instead of technics

Response: It has been corrected

**Reviewer2 :**

I have reviewed "Modeling subgrid lake energy balance in ORCHIDEE terrestrial scheme using the FLake lake model". In the manuscript, the authors indicate the importance of including lakes in ESM and also do a first attempt. I think the approach used by the authors is used before (e.g. the VIC model also includes lakes for the same reason), however, the approach is novel as it uses several lake depths ranges to represent the distribution of lakes over the different depths.

Response: We would like to thank Reviewer 2 for taking the time to read and thoroughly examine our manuscript. Your thoughtful comments have helped us to improve and strengthen our paper.

The manuscript is well-written, and I have only comments on some clarity points especially I suggest to critically look at the clarity of the figures:

L41 "Regarding the carbon and nitrogen cycles, lakes emissions are poorly constrained but it is recognized that their contribution is significant". --> needs a reference

Response:  We added the following reference : Methane emissions from lakes: Dependence of lake characteristics, two regional assessments, and a global estimate, David Bastviken, Jonathan Cole, Michael Pace, Lars Tranvik, 2004.

L80: "and various developments have been done to assess the lakes features and set up the new parameterizations" --> I suggest making this a bit more clear on what these various developments include

Response: The sentence has been revised

L103: "for the following developments" --> what developments do the authors mean here?

Response: sorry for the ambiguous sentence: we were meaning future developments concerning the modeling of the water budget of lakes and the connection to the river routing module. We modified this sentence in the revised manuscript to avoid confusion, into: "This specificity will insure the consistency of our future developments around the modeling of the lake water budgets and the lake-river interactions".

L126: "The data are available for some lakes back to the year 1850 (Benson, 2002)." And till when does this dataset run? Is it daily updated?

Response: Sorry, the dataset was not presented in sufficient details: the time series are very heterogeneous among the lakes, most of them cover a period of 20 years and some can reach more than 100 years. The database has been updated in 2020 and includes data up to this year. We have modified in the revised version the presentation of the database and especially described how the ice-on/off and duration were defined, in order to better understand the model-observations discrepancies.

L128: "Different atmospheric reanalysis are used to force the ORCHIDEE model and assess the model sensitivity to meteorological forcing uncertainties." Do the authors mean in this

study or by others. In the case of this study, some more details are required. In case it is by others, I suggest adding a reference.

Response: Yes, we were meaning this work, sorry for the error in the tense of this sentence. We have corrected it.

L161 "This procedure is fully presented in Lurton et al. (2020) and more recently in Harper et al., in preparation" Given that Harper at all presents the procedure again in a yet unpublished paper, I was wondering if anything changed in the procedure. And if so, which version was used in this study?

Response: no the procedure did not change and is fully presented in Lurton et al.,2020. We have removed Harper's reference, since it is not published yet.

L168: NWP is only used twice, so for clarification perhaps it can be just fully written here.

Response: done

L184 "FLake does not model the hypolimnion, the layer under the thermocline which is present or may appear seasonally in stratified lakes and where the water density is the highest with a constant temperature around 4 C" What do the authors mean with constant temperature around 4C? The hypolimnion can be in some cases several degrees deviating from 4 degrees Celcius as a quick search on the internet shows.

Response: sorry, the sentence has been modified, the hypolimnion, unlike the thermocline, is quite isolated and presents a temperature close to 4°C in deep temperate lakes, but can be much larger in tropical regions. The sentence has been corrected to be more precise.

L242 "..but given that the use of on a constant value " --> it seems that the word 'on' should be deleted

Response: agree

Figure 2 needs some more clarification on what the different panels show (now only fractions and depths are provided, but I do not see 2 graphs but 8). Moreover, I suggest adding the word median depth to the caption to indicate that the values for depth are median and not mean values.

Response: we have completely redone this figure following also the other reviewers suggestions.

Paragraph 3.4: do the authors mean with lake tiles the 3 lake depth categories: >5m, 5-25m, and <25m? If so I suggest referring to this in paragraph 3.4 to clarify. (later in the manuscript I see this is indeed the case, but at this point this needs clarification)

Response: we agree, we have revised this paragraph following your advices.

L303 "4.1.1 LWST" This abbreviation needs to be written fully

Response: It was a typo, we meant LSWT which is defined at the beginning of the document in line 114.

L309 "This sampling is representative of the area lake distribution plotted in Figure 2" It is not so clear how I can see this. Related to this, I was wondering if figure 3 thus compares representative lakes for a certain grid with real lakes? And how is this done? By comparing a real lake with the tile it falls in?

Response: we wanted to say that the latitudinal/longitudinal distributions of lakes are similar to the lake fractions derived from HydroLAKES database which were also plotted in Messager et al.,'s paper. We revised this sentence in the manuscript. For the comparison simulation/observations, yes you are correct, we compare the real lakes to the simulated tile it falls in the ORCHIDEE grid. We modified the text to make it more clear.

Figure 3 shows on top and to the right two scales, one for the number of lakes and one for the RMSE. However, it is not clear which scale is for which variable. I can only hope that 0-75 is not for the RMSE, but this is not indicated.

Response: there was an error in the legend, it has been corrected in the new version of the figure.

L332: what are "height lakes"? I first thought of mountainous lakes, but Lake Tai is definitely not a mountainous lake.

Response: sorry for the mistake, we meant eight lakes instead !

Figure 5. The excess of especially the scatter plots needs more explanation.

I think the prescribed step described at L395 is more for the methods section.

Response: we have moved this part in the methods section following your advices.

**Reviewer 3:**

Reviewer's comments on

Modeling subgrid lake energy balance in ORCHIDEE terrestrial scheme using the FLake lake model by Bernus Anthony and Ottlé Catherine. This study presents new prognostic lake parametrizations in the ORCHIDEE surface modelling system within the global IPSL earth system model. Parametrizations of in-lake variables are based on Freshwater Lake (FLake model). Simulated lake water temperatures and ice phenology are compared against global observational data sets. The resolution of the atmospheric forcing, that is derived from various reanalyses, is shown to play important role for the simulated lake surface energy balance. It was found that lakes generally freeze earlier and melt later in the model than according to the observations. Uncertainties in modelling of freezing and melting of lake ice are discussed. Seasonal LWST distribution was found to be realistic in the majority of lakes studied. The manuscript reports results of extensive modelling and intercomparison work in a systematic way. The results mostly confirm earlier findings reported in literature during the last more than ten years. The authors are well aware of the previous studies related to the implementation of FLake parametrizations into different NWP and climate models and build their system taking this experience into account. An idea of treating lakes of different depth using prescribed separate subgrid tiles is suggested. The essential things are presented and discussed, valid methods used and the results seem reliable. Use of external data sets is well documented and references given.

A problem of the study may be that it covers too much areas, topics, data, models in the global scale and tries to reach conclusions on everything.

Would it be possible to discuss shortly the specifics of lakes in a global land-surface models v.s. high-resolution regional weather models?

Response: We would like first to thank the reviewer for his/her constructive comments. We have revised our manuscript following your suggestions.

Concerning your comment on the need to focus on specific lakes, we would like to note that our purpose is the development of a lake modeling at global scale in a land surface module included in a global earth system model. Therefore, we focused the evaluation at a global scale. At the scale of the model grid (typically 0.5°) few lakes can be explicitly represented and most of them are aggregated in a generic lake tile with effective parameters resulting from the agregation. In the evaluation also, we focused on the variables that matter for the ESM energy budget, therefore lake surface temperature and energy fluxes as well as freezing states. Compared to high resolution models that require a better spatial discretization to model precisely the lake-atmosphere interactions on the atmosphere dynamics, for example the regional lake impacts on the energy and humidity horizontal and vertical advection, that are crucial to model and predict the regional weather and climate. Here at such large scale, the vertical processes are more important and it is crucial to correctly represent the temperature and humidity vertical gradients.

We highlighted this point in the introduction section, see lines 52-56.

FLake has been applied in both, quite a lot of reports exist in the literature about the NWP-model side results. Unfortunately the manuscript leaves a somewhat raw impression. It is not always easy to follow the reasoning of the authors, clarifications would be needed in several places. It would be good for the authors to go carefully through the whole text in order to improve the presentation, correct typos and English language mistakes. Some comments are given below in the specific remarks.

Response: the manuscript has been thoroughly revised following all the comments and suggestions of reviewers, all the figures were redone, and we have tried to clarify all the unclear sentences/sections highlighted by the reviewers.

Specific remarks

Abstract
* * *
l. 13-14 Which one of CRUJRA or ERA5 has smaller RMSE error ("best and worst")? c.f. l. 369-370.

Response: ERA5 presents the smallest errors. We have revised the sentence to make it more clear.

2. Data and modelling framework

l. 92 Please define PFTs here when using the acronym the first time

Response: "PFTs" stands for "Plant Functional Types (PFTs)", we have defined it here.

l. 97- Could you please comment the use and differences between HydroLakes and Global Lake Data Base (GLDB), which is the native lake depth data base for FLake.

Response: Kourzeneva et al., 2010 developed GLDB for ECOCLIMAP context. Since we dont use the ECOCLIMAP land cover maps, and more over use the Hydrosheds data for river network definition, it was more convenient for us to use Hydrolakes.

l. 142-243. The last sentence is unclear, please clarify.

"In brief, the interpolation of the meteorological variables is done linearly except for the solar radiation where the interpolation between two times, follows the typical daily evolution according to the solar zenithal angle. It should be noted that no extra atmospheric fields are needed to solve the energy balance at the lake – atmosphere interface, same as what is done for the vegetation part." → ?????????

Response: we wanted to say that the same atmospheric variables are driving the energy balance on the vegetation and on the lake areas within the grid cell. The sentence was clarified in the revised version, a reference was added also. see lines 147 - 150.

l. 180 Please tell how the surface extent is used in FLake within the ORCHIDEE grid.

Response: The lake surface extent is used in to estimate the lake fraction within the ORCHIDEE grid cell and also to estimate the fetch parameter, but this is done in a preprocessing phase. It is no more needed to run the model, therefore, we have modified this sentence, the average depth and radiative parameters of the lake tile are the only input parameters of FLake in our configuration.

3. ORCHIDEE-FLake coupling developments

l. 204 typo: parametrized

Response: It has been corrected

l. 276 Lake Baikal (i, not the French double i)

Response: It has been corrected at each occurence

l. 282-283 Please clarify the circular shape

Response: The fetch parameter has been calculated for each lake of the tile according to its surface extent, assuming that all lakes have a circular pattern. The fetch tile value was assessed by averaging all the individual values weighted by their surface extent.

Therefore, we changed the sentence : "Considering a circular pattern for the lake shape" into: "Assuming a circular pattern, the fetch was set to the diameter of a circle of equivalent surface, and the effective fetch of the tile was derived from the aggregation of the fetch values of all the lakes falling in the lake tile considered (mean average)."

l. 288 What exactly means configuration without lakes - what kind of surface was assumed instead of lakes?

Response: In the ORCHIDEE standard version, water surfaces are treated as bare soils (this was explained in section 2.1.1). Wwe added to specify "(i.e. considered as bare soil)"

l. 291-92 What means 20 years of spin-up? Since 1980?

Response: To equilibrate the model prognostic variables, we run ORCHIDEE for a number of successive years before being able to get reliable simulations. In our case, we started the simulations on year 1979 which is the first year available in some of the forcings, and analyze the simulations from the year 2000. We have modified the sentence to be more clear and precise.

l. 294 How many characters? Four or perhaps 3-5?

Response: Sorry, we meant 3-5 letters for the way of depth agregation. It has been changed in the text.

**4. Results**
* * *
l. 321 How did you derive SOF and EOF from the model simulation results? Based on simulated ice thickeness, LWST? Please specify.

Response: Ice on/off dates were derived from the surface temperature simulated time series. We have specified it in the revision and discussed it more deeply in the manuscript, since this choice may explain part of the errors.

l. 331 typo eight (not height)

Response: "h" has been removed

l. 339 mistake: variations ... are

Response: corrected

l. 361-362. Please clarify the first sentence. What means correct depth class tile here?

Response: we meant the tile in which the lake falls in. We have corrected the sentence to be more clear.

**5. Discussion and conclusions**

l. 493-496 Is this an attempt to evaluate the reanalysis results?

Response: No, comparison and evaluation of reanalysis datasets have been the subject of numerous studies and papers and in our case, we can not say that better agreement of our lake temperatures with observations is explained by the better quality of some of the forcings. By comparing various atmospheric datasets, we expected to separate model errors from forcing errors and identify systematic biases. This is what is discussed in the next paragraph, and this sentence was slightly modified in the revised version.

Perhaps it would be better to refer to their own considerations instead of suggesting simplified interpretations.

Response: We are not sure to understand if this comment is the rest of the previous sentence or another comment referring to another part of the discussion ? Anyway, the discussion and conclusion parts have been thoroughly revised and we dont think that we are suggesting simplified interpretations. Sorry if we missed something.

**Reviewer 4**

Summary

This manuscript reports on experiments with the 1-D lake model FLake, which has been incorporated into the ORCHIDEE land surface model for global scale simulations driven by various atmospheric forcing datasets. This is the first (and necessary) step in the development of a fully coupled atmosphere-lake modelling system (in this case LMDZ apparently). Even though FLake has already been coupled with other NWP systems, and some of the deficiencies found here are already known, this study is still of interest because the problem of lakes in environmental prediction is so important. However, as outlined below, some clarification is needed in a few areas, and interpretation of some of the results seems a bit oversimplified. Also, the paper would be strengthened with a more detailed analysis of bias, and some discussion on the problem of partial ice cover.

Response: We would like first to thank you for your comments and suggestions. The manuscript was thoroughly revised following the four reviews and especially the evaluation of the ice cover is now discussed more deeply, the observations are better presented and the problem of the partial ice coverage is better discussed.

Key Points

The 1-D lake model descriptions (especially how they differ from each other) and references mentioned in the Introduction could be improved. In FLake only the metalimnion temperature profile is estimated via a similarity argument, not the entire column (Line 57). In fact a surface mixed layer is computed based on a (more or less conventional) bulk turbulence kinetic energy (TKE) approach. As such the model has only two layers: a surface mixed layer (with a computed mean temperature and thickness), and a metalimnion (with parameterized temperature profile), making it exceptionally inexpensive to run in terms of computer resources. In contrast, the Hostetler model solves turbulent mixing on a grid (*e.g.* 10 – 25 levels) based on a parameterization of eddy diffusivity. An important reference for this approach, and also the model LISS, that should be mentioned is Subin *et al.*, 2012 (*JAMES*). I agree that in some sense the CSLM (MacKay, 2012) is a kind of "hybrid" approach – employing a similar bulk TKE surface mixed layer model to that used in FLake, along with a resolved metalimnion and hypolimnion (~ 20 levels). A very recent CSLM study – one quite similar to your own – is Garnaud *et al.* 2022, *JAMES*. Because the goals of this study have significant overlap with your own, a comparison of your results with theirs would be of great interest. Note that the CSLM was not included in the LakeMIP studies of Stepanenko et al. (Line 60). On the other hand, that study did include several k – ε models, representing an alternative to the simple Hostetler eddy diffusivity approach of modelling turbulent mixing. If any of these has been coupled to a land surface or atmospheric model it might be worth a mention in your Introduction.

Response: The model LISSS was mentioned without the reference, we add the associated reference as you recommended it. We discuss the result of Garnaud et al. 2022 in the discussion part.  We corrected the mistake where we mentioned that CSLM was included in

the LakeMIP study . We did not added k - ε models since they have not been used in ESMs, and this introduction was focused on lake modeling in ESMs.

The LSWT RMSE seems fair in general (2 – 4 K), but how do these values compare with other studies?

Response: The comparison with the results obtained by previous studies are discussed in the manuscript. We only focused on the global scale works and forced simulations, which were presented in Dutra et al. 2010. We have more difficulties to compare with others studies such as Lemoigne et al. 2016 where only the difference of RMSE between the different experiences are shown.

Also, there is too little discussion of bias. The ice phenology analysis needs a bit more clarity. It seems that the start of freezing (SOF) indicator – both in the simulation and the observed data – refers to the date of complete (*i.e.* 100%) ice cover. Since FLake cannot really model fractional ice cover this seems reasonable. But many large lakes, for example the Laurentian Great Lakes, virtually never freeze over completely though they all produce partial ice cover. Thus there may be a large error in the simulation of ice cover that you are missing. Your bias statistics in Table 2 can only include lakes that freeze completely in both the simulation and the observations (in order to compute the bias), and thus under-represent a potentially important model deficiency in this (and all) 1-D lake models. This is an important point and warrants some discussion.

Response: We only take lakes that are completely frozen in both simulations and observations. We highlight this point in our new version according to your remark (line 340-342). We discussed more deeply about this deficiency in the discussion part in this new version.

It is interesting to see the partition of RMSE into bias and other components in Fig. 4 but the analysis is incomplete in that the sign of the bias is not analyzed. The 8 lakes examined in the figure seem to generally show a positive LSWT bias – is that correct? Is that the trend globally? If so that could point to a systematic error in FLake. A hint is shown in Fig. 5 and in the second last paragraph of §4.2.2, but it would be very interesting to show bias results in a figure like Fig. 3, for example, or some kind of table.

Response: YES we obtain a systematic positive bias in our simulations, and this is shown in the MSE decomposition results and in the SB component. The interpretation of these biases is not straigthforward since it varies a lot among the different lakes and climates. Depth differences (between the observed lake and the modeled tile) seem to explain most of the errors for some of the specific lakes we have examined more deeply. Some pathways to better understand and possibly improve these errors are given in the discussion part, and require future work.

It is somewhat oversimplified to state that the lake freezing process is determined by lake depth (Line 507);

Response: this is the case in FLake (even though the fetch parameter depends on surface area in our case) and it was what we wanted to say. But we agree that in the real world, the lake extent and wind stress are also very important parameters to take into account.

lake surface area is an important factor in ice mechanics (*e.g.* Leppäranta and Wang, 2008, *Hydrobiologia*). For example, the mean depth of Lake Erie is only about 19 m but it rarely freezes solid. You also suggest (§5.2 second paragraph) that snow conductivity and transport might be important processes for EOF results – what about snow-ice production? Is all of the ice produced by FLake through congelation (*i.e.* black-ice) processes? Note that even a "perfect" snow model can always be tuned to improve your ice phenology statistics (*e.g.* reducing albedo to accelerate ice-off). I think it is preferable to consider missing processes before adjusting physical constants whenever possible.

Response: we agree, and try to better explain these missing processes in the discussion part, see lines 549 - 555.

Minor Comments

- Fig.3 caption differs from annotation: I assume RMSE is red, as stated in caption

Response: YES, the figure was corrected, sorry for this mistake

- Fig. 6 caption states RMSE but the figure seems to show bias

Response: what has been plotted is the dispersion of the bias among all the lakes observed, we corrected the legend in the revised version.

- Line 516: you state that SOF corresponds to the first occurrence of ice but elsewhere in the paper, as well as in the documentation for the observations (https://nsidc.org/data/G01377/versions/1) SOF corresponds to the first day of complete ice cover. Please clarify. This is an important distinction – the ice-on process *i.e.* the period from first appearance of ice until 100% ice cover can last many weeks for larger lakes (with some lakes never freezing completely at all). This period is not analyzed at all in your study.

Response: we agree that because FLake do not simulate the partial ice coverage, the first occurence of ice in FLake is probably in advance for many lakes compared to the observations that report only the complete ice coverage. We discussed this point more deeply in the discussion part and proposed ways of improvement to simulate this process in the following.

---

## Referee Report (RR1)

Review of "Modeling subgrid lake energy balance in ORCHIDEE terrestrial scheme using the FLake lake model" by Bernus Anthony and Ottlé Catherine

Article shows results of testing 1-D lake parametrization scheme FLake incorporated in the ORCHIDEE land surface scheme. Experiments are performed at different horizontal resolutions and temporal scales with different lake mean depth aggregation techniques. Experiment results are compared with observed lake surface temperatures and ice start, end, duration dates.

GENERAL COMMENTS

In the paper lake depth from HydroLAKES is used as ground truth, yet over some latitudes and regions this data plays crucial role. It would be good if data was crosschecked/verified at least for categories where only few lakes were used (small number of lakes was used).

For different lake categories errors are provided, but it is obvious that not all categories have vast amount of lakes inside, so it would be good to see if numbers presented have statistical significance. For example errors can be high for some latitudes but are based only on two lakes, so this is statistically not significant at all.

Authors use different lake mean depth aggregation techniques – it would be a great advantage if they could also add MODE aggregation technique, or at least say about it in discussion.

Paper has huge amount of abbreviations and some are not explained (e.g., line 74 – LM4, GFDL, etc.) – it would be a big help for the reader to have as little abbreviations as possible, e.g. to keep only the ones that are used all the time, like LSWT, add remove all unnecessary ones, e.g. write them in full and add abbreviation in brackets if necessary.

In Section 4.2.1 model is evaluated based on 8 lakes, but all these lakes are huge and their surface temperature can be also due to internal currents, wind fetch, etc., so these lakes can't represent lakes globally even though they are located in different climate zones. Most of lakes globally are quite small in area and around 10 m deep (mean depth). Results are interesting and well presented but it should be noted that they are not representative globally.

Some sections, like Section 4.3, have a very interesting results, for better understanding it would be great to put it in a table – all numbers easy to compare and understand.

TECHNICAL COMMENTS

l.31 "Lake distribution is spatially unequal all other the world with two regions …" – should instead of "other" be "around"?

l.69 "the UK Met Office Unified Model and its JULES Land simulator (Rooney and Bornemann, 2013)" – I guess FLake there is not couples operationally yet.

l.114 (same as l.31) – "all other the world" – should it be "around"?

l.183 "the free water bottom temperature" – what is meant here exactly?

l.243 "to adjust the time split factor which has been finally set to a value of 50" – could you explain with more details?

l.387 (same as l.31) – "all other the world" – should it be "around"?

Figure 4 – explain please column plots – what different shading represent?

Figure 7 – explain please left plot – what columns and lines represent exactly?

l.589 "GHG fluxes like carbon and methane…" – did you mean carbon dioxide?

---

## Author Response (AR2)

**Second review**

**Reviewer 1**

I would like to thank the authors for their revised manuscript, which provides answers to questions raised during the initial review process.

We want to thank reviewer 1 to revise the first and our second version of our manuscript. We follow your last technical comments to do the last correction.

Technical Comments:

L68: replace (Stepanenko et al., 2010, 2013) by Stepanenko et al. (2010, 2013)

L127: replace we use the period 2000-2016 by the period 2000-2016 was used

L130&131: replace day time by daytime and night time by nighttime

L195: replace and that the water balance is not solved by and not the water balance

L197: I suggest to remove the parenthesis and add comas because albedo and extinction coefficient are as important as fetch

L262: 2.29 W m$^{-1}$ K$^{-1}$

L312: remove the extra space after soils

L316: it is not clear if the spin up was run before 1979 or if it corresponds to the period 1979-1999 (which seems to be the case). If it is so then you could replace "A spin up procedure of 20 years in the previous years was applied to bring…" by "The first 20 years (1979-1999) were used as spin up to bring..."

L327: replace succeed by succeeded

L328: replace are by were

L330: replace correspond and assign by corresponded and assigned

L408: replace Erie and Superior lakes by Lake Erie and Lake Superior

L410: replace Baikal lake by Lake Baikal

L428: replace It should be noted also by It should also be noted

L470: add unit after -0.7

L473&474: show is used 4 times, please rephrase

L463: the meaning of the curves (solid lines) in the description of Figure 7 (histogram) is missing

L589: evaluation can be removed (Choulga et al. showed the impact of depth on ice phenology)

L593: replace do LSWT by does LSWT (meaning surface temperature in general)

**Reviewer 2**

Review of "Modeling subgrid lake energy balance in ORCHIDEE terrestrial scheme using the FLake lake model" by Bernus Anthony and Ottlé Catherine Article shows results of testing 1-D lake parametrization scheme FLake incorporated in the ORCHIDEE land surface scheme. Experiments are performed at different horizontal resolutions and temporal scales with different lake mean depth aggregation techniques. Experiment results are compared with observed lake surface temperatures and ice start, end, duration dates.

We want to thank reviewer 2 for its useful comments.

**GENERAL COMMENTS**

In the paper lake depth from HydroLAKES is used as ground truth, yet over some latitudes and regions this data plays a crucial role. It would be good if data was crosschecked/verified at least for categories where only few lakes were used (small number of lakes was used).

Depth validation has been done on a large amount of lakes (5101 lakes) during the elaboration of the HYDROLAKES database and the authors also provide an estimation of the error on the depth. For the 8 large lakes highlighted in the local scale study, we have checked that the HYDROLAKES depth as well as the tile depth simulated are well in accordance with the published literature on the subject. To account for depth uncertainty, we could generate different depth maps for our modeled tiles and look at the impact on the model results but this is another study and could be also the subject of a next paper.

For different lake categories errors are provided, but it is obvious that not all categories have vast amount of lakes inside, so it would be good to see if numbers presented have statistical significance. For example errors can be high for some latitudes but are based only on two lakes, so this is statistically not significant at all.

You are right, and it is for this reason that we do not compare the RMSEs of each category of lake type. In our comparisons (impacts of depth parameterization of atmospheric forcings) , we always compare statistics performed on the same number of lakes. In Figure 3, since indeed the number of lakes vary with latitude and longitude, we have plotted the number of lakes which have been included in the comparison, to put in perspective our results.

Authors use different lake mean depth aggregation techniques – it would be a great advantage if they could also add MODE aggregation technique, or at least say about it in discussion.

The mode value of the lake depths within a tile could be also a way to define the tile depth and will give more weight to the main peak of the distribution if any. We have seen that in some cases where the lake distribution is very heterogeneous, this could lead to significant differences compared to the mean or median values. Furthermore, a large peak mode could mix up shallow and intermediate lakes presenting different physical behaviors and it is preferable to differentiate them.  In this study, we wanted to define tiles with depths representative of the 0.5° grid cell, so we think that the median or the mean is a better approach than the mode, and this is why we did not include the mode approach in this work.

Paper has huge amount of abbreviations and some are not explained (e.g., line 74 – LM4, GFDL, etc.) – it would be a big help for the reader to have as little abbreviations as possible, e.g. to keep only the ones that are used all the time, like LSWT, add remove all unnecessary ones, e.g. write them in full and add abbreviation in brackets if necessary.

We removed all the unnecessary abbreviations and checked that all were well defined. We agree that in the presentation of the state of the art, there are a huge amount of acronyms but we think that it is important to leave them because the models in our community are more known by their acronyms than by their full names. Therefore, we don't see any other satisfactory solution than leaving the model names, but we entirely write the institute names instead of their acronyms to limit them at best and ease the reading.

In Section 4.2.1 model is evaluated based on 8 lakes, but all these lakes are huge and their surface temperature can be also due to internal currents, wind fetch, etc., so these lakes can't represent lakes globally even though they are located in different climate zones. Most of lakes globally are quite small in area and around 10 m deep (mean depth). Results are interesting and well presented but it should be noted that they are not representative globally.

We added a sentence to notice this fact in the corresponding section.

Some sections, like Section 4.3, have a very interesting results, for better understanding it would be great to put it in a table – all numbers easy to compare and understand.

We added a table which gives the 5th and 9th decile of the depth error distribution and which is complementary to figure 7. Thanks for your suggestion.

TECHNICAL COMMENTS

l.31 "Lake distribution is spatially unequal all other the world with two regions …" – should instead of "other" be "around"?

Done

l.69 "the UK Met Office Unified Model and its JULES Land simulator (Rooney and Bornemann, 2013)" – I guess FLake there is not couples operationally yet.

l.114 (same as l.31) – "all other the world" – should it be "around"?

Done

l.183 "the free water bottom temperature" – what is meant here exactly?

Done

l.243 "to adjust the time split factor which has been finally set to a value of 50" – could you explain with more details?

We added in this paragraph that we made some tests for different values of this split factor and explained that we had to increase this factor to a value of 50 to remove all the instabilities. We hope that this part is better explained now.

l.387 (same as l.31) – "all other the world" – should it be "around"?

Done

Figure 4 – explain please column plots – what different shading represent?

Done

Figure 7 – explain please left plot – what columns and lines represent exactly?

We added in the label of the figure that the lines represent the smoothed distributions.

l.589 "GHG fluxes like carbon and methane…" – did you mean carbon dioxide?

Done